# The association of developmental trajectories of adolescent mental health with early-adult functioning

Anoek M. Oerlemans◉*, Klaas J. Wardenaar, Dennis Raven, Catharina A. Hartman, Johan Ormel

Department of Psychiatry, Interdisciplinary Center Psychopathology and Emotion Regulation (ICPE), University of Groningen, University Medical Center Groningen, Groningen, The Netherlands

* a.m.sluiter-oerlemans@umcg.nl

**Data Availability Statement:** Under the GDPR, our dataset is considered pseudonymized rather than anonymized, and is still regarded as personal data. Given that participants have not given informed

## Abstract

### Background

Mental health problems during adolescence may create a problematic start into adulthood for affected individuals. Usually, categorical indicators of adolescent mental health issues (yes/no psychiatric disorder) are used in studies into long-term functional outcomes. This however does not take into account the full spectrum of mental health, nor does it consider the trajectory of mental health problem development over time. The aim of this study was twofold: (1) to identify distinct developmental trajectories of (co-occurring) internalizing and externalizing mental health symptoms over the course of adolescence (ages 11–19), and (2) to document the associations between these adolescent trajectories and economic, social, and health outcomes in young adulthood (age 22), unadjusted and adjusted for childhood functioning, putative confounders and current mental health.

### Methods

Data were used from the Dutch TRAILS cohort study (subsample n = 1524, 47.3% males). Self-reported INT and EXT symptoms using the Youth/Adult Self Report were assessed four times (ages 11y, 13y, 16y, 19y). Adolescent mental health trajectories were estimated using Parallel-Processes Latent Class Growth Analyses. Self-reported economic, social, and health outcomes and parent-reported current mental health (using Adult Behaviour Checklist) were assessed at age 22. Multiple logistic regression analyses were performed to test associations between trajectories and outcomes.

### Results

Four distinct trajectory classes were identified: (1) a normative class with decreasing-low INT+EXT symptoms (n = 460), (2) continuous moderately-high INT+EXT (n = 298), (3) continuous moderate, INT>EXT (n = 414), and (4) decreasing moderate, EXT>INT (n = 352). Compared to the normative class, the other three trajectories generally predicted less optimal early-adult outcomes, with the strongest effects observed for individuals with continuous

consent to have their personal data publicly shared, we are legally and ethically not allowed to publicly post our dataset. Except for a contribution to the handling coasts of 2500 euro, these data are available free of charge, but not freely accessible; access can be obtained by submitting a publication proposal. The conditions imposed on the access to and use of TRAILS data are laid down in the general conditions (version January 2019; only in Dutch; accessible through https://www.trails.nl/en/hoofdmenu/data/data-use) and the TRAILS general conditions data use for external users (version October 2015, https://www.trails.nl/en/hoofdmenu/data/data-use). These describe the procedure for requesting data, the demands placed on the manuscript, and the demands imposed on the use of the data. In general, TRAILS data are available for scientific researchers affiliated with TRAILS (internal consumers) and scientific researchers in the medical and social sciences in general (external consumers). To apply for TRAILS data by external users, the 'publication plan TRAILS – external users' form has to be completed (available from: https://www.trails.nl/hoofdmenu/data/data-use) and submitted to the TRAILS secretariat (e-mail address listed at the bottom of the form) for approval by the TRAILS management team. Only the data specified in approved proposals are distributed by the data manager. All data users who fail to comply with the TRAILS access regulations will not receive approval of their publication plan. Any data request without such approval will be denied. After approval, data can be obtained from the data manager. These procedures have been in use for many years. For more information and questions, please contact the scientific coordinator Prof. Dr. Judith Rosmalen (j.g.m.rosmalen@umcg.nl). Codebooks are available from DANS EASY, the online archiving system for research data by the Netherlands Institute for Permanent Access to Digital Research Resources (https://easy.dans.knaw.nl/ui/?wicket:bookmarkablePage=:nl.knaw.dans.easy.web.search.pages.PublicSearchResultPage&q=Tracking+adolescents%27+individual+lives+survey). We confirm that the TRAILS data and codebooks available from DANS EASY are the minimal data set underlying our study. The variables included in our analysis are available from DANS EASY, except for the data from the Psychiatric Case Registry – North Netherlands which was, amongst others, used to determine lifetime mental health care use. Access to these data is available for researchers who meet the criteria for access to confidential data (on request through the 'publication plan TRAILS – external users' form). To view the TRAILS codebooks, you need to login to the DANS EASY

moderate-high levels of both INT and EXT symptoms throughout adolescence. The associations largely remained after adjustment for pre-adolescent functioning, selected confounders and current mental health.

## Conclusions

Both adolescent trajectories and current mental health had substantial independent effects on early-adult functioning.

## Introduction

Childhood and adolescence are critical periods in terms of the pathogenesis of mental health problems. The first ever onset of mental disorders often occurs in childhood and adolescence and the lifetime prevalence of diagnosable mental disorders increases substantially during adolescence [1,2]. Among youths aged 10 to 24 years, mental disorders are the leading contributor to the burden of disease [3]. Some portion of this burden is due to relatively rare (<1%) but very impairing chronic mental disorders (e.g., schizophrenia and autism spectrum disorder). Most of the burden, however, is due to common mental disorders and subclinical mental health problems as documented by follow-up studies e.g. [4–7]. These mental health problems may create a problematic start into adulthood for affected individuals, while adequate functioning during the transition into adulthood may be crucial given the developmental challenges that lie ahead, such as completing one's education, getting and maintaining a job, avoiding illegal and health risk behaviors, and developing and maintaining a social support network [8,9]. In the current study, we report on how common mental health problems during late childhood and adolescence affect functional outcomes in early-adulthood.

### A longitudinal approach: Developmental trajectories

Several studies have looked at the long-term functional outcomes of individuals with childhood and adolescent psychopathology [see e.g., 9–12] and have thus far indicated that having any type of mental disorder predicts adverse outcomes related to health behaviour, social functioning, and legal and financial issues. Also in the current sample, we found associations between childhood psychiatric diagnosis and a wide range of adverse economic, social, psychological and health-behaviour outcomes [13]. Most of these studies, including our previous study, have defined adolescent psychopathology in terms of DSM-IV diagnostic categories (i.e., yes/no disorder). However, studying only adolescents who meet full criteria for mental disorders may underestimate the burden. Children and adolescents with symptoms that do not meet the criteria for a DSM diagnosis may still suffer from impairment and risk of continuing problems and future disorders as a result of those symptoms [8]. Therefore, a more complete understanding of the impact of adolescent mental health issues on early-adult functioning should not be limited to DSM disorders, but requires the inclusion of the full spectrum of mental health, including subclinical symptoms. In their 2015 study, Copeland and colleagues included subthreshold cases (defined as individuals with significant impairment resulting from any psychiatric problems that do not meet full DSM criteria for a disorder) and provided evidence that childhood psychiatric problems are associated with a disrupted transition to adulthood even if the problems are subthreshold [9]. Although the inclusion of such a categorical subthreshold group was already an important improvement in understanding the

website. You can do that by creating a (free of charge) EASY account or login through your institutional account. URLs and DOIs for the five measurement waves (T1-T5) of the population cohort of TRAILS are provided below: TRAILS T1 population cohort; URL: https://easy.dans.knaw.nl/ui/datasets/id/easy-dataset:33750, DOI: 10.17026/dans-xw6-72fh TRAILS T2 population cohort; URL: https://easy.dans.knaw.nl/ui/datasets/id/easy-dataset:36491, DOI: 10.17026/dans-x7s-svmu TRAILS T3 population cohort: https://easy.dans.knaw.nl/ui/datasets/id/easy-dataset:52566, DOI: 10.17026/dans-27s-27kw TRAILS T4 population cohort; URL: https://easy.dans.knaw.nl/ui/datasets/id/easy-dataset:62108, DOI: 10.17026/dans-xnb-mbh9 TRAILS T5 population cohort; URL: https://easy.dans.knaw.nl/ui/datasets/id/easy-dataset:62111, DOI: 10.17026/dans-2aa-wj84

**Funding:** The authors received no specific funding for this work. However, this research is part of the TRacking Adolescents' Individual Lives Survey (TRAILS). TRAILS has been financially supported by various grants from the Netherlands Organization for Scientific Research NWO (https://www.nwo.nl/en/) (Medical Research Council program grant GB-MW 940-38-011; ZonMW Brainpower grant 100-001-004; ZonMw Risk Behavior and Dependence grants 60-60600-97-118; ZonMw Culture and Health grant 261-98-710; Social Sciences Council medium-sized investment grants GB-MaGW 480-01-006 and GB-MaGW 480-07-001; Social Sciences Council project grants GB-MaGW 452-04-314 and GB-MaGW 452-06-004; NWO large-sized investment grant 175.010.2003.005; NWO Longitudinal Survey and Panel Funding 481-08-013 and 481-11-001; NWO Vici 016.130.002 and 453-16-007/2735; NWO Gravitation 024.001.003), the Dutch Ministry of Justice (WODC, https://english.wodc.nl/), the European Science Foundation (http://www.esf.org/) (EuroSTRESS project FP-006), the European Research Council (https://erc.europa.eu/) (ERC-2017-STG-757364 and ERC-CoG-2015-681466), Biobanking and Biomolecular Resources Research Infrastructure BBMRI-NL (https://www.bbmri.nl/) (CP 32), the Gratama foundation (https://www.rug.nl/alumni/support-research-and-education/groninger-university-fund/gratama-stichting/) (2017/227), the Jan Dekker foundation, the participating universities (University Medical Center and University of Groningen, the University of Utrecht, the Radboud Medical Center Nijmegen, and the Parnassia Group), and Accare Centre for Child and Adolescent Psychiatry (https://www.accare.nl/). The funders had no role in study design, data collection and analysis, decision to publish, or preparation of the manuscript.

impact of (a broader spectrum of) mental health issues on early-adult functioning, it did not take into account the developmental trajectory of mental health problems over time nor did it consider how specific mental health symptoms co-develop or change (dis)concurrently over time within an individual. Individuals were regarded as subthreshold when they met criteria for symptomatic impairment during any of the six assessments in childhood and adolescence (between ages 9–16 years), irrespective of age of onset, type of problem or comorbidity. Yet, symptoms with an early onset of development may have a worse prognosis in terms of long-term consequences than symptoms that developed later in life because of prolonged rippling effects on the way a person masters key educational, occupational or social challenges. This hypothesis is supported for example by the finding that earlier ages of onset of depression were associated with substantial functional impairment, and greater illness burden than later ages of onset [14]. Although, in our 2017 study, we found the opposite pattern of individuals with adolescent-onset psychiatric disorder being more likely to feel lonely or smoke compared with individuals with childhood-onset disorders [13], suggesting that more research is needed to better understand how different onsets and trajectories of mental health symptom development may differentially affect functional outcomes.

## Combining trajectories of internalizing and externalizing problems

Moreover, some specific symptoms or combination of symptoms from one or more symptom domains (i.e., INT, EXT) or pattern of development may have a worse prognosis than others and may be associated with adverse outcomes in different ways. Thus, individuals with a developmental trajectory of e.g. early-onset anxiety problems only may have a different set of associated functional outcomes compared with individuals with a developmental trajectory of early-onset behavioural problems + later-onset mood problems or individuals with later-onset comorbid mood + anxiety problems. So far, only a few studies have examined trajectories of INT and EXT symptoms using growth mixture models and latent class growth models [11,12,15]. Although the number of identified subgroups of individuals with relatively similar trajectory patterns differs between these studies, commonly found classes of trajectories include two or three relatively stable trajectories that differ in initial problem level (low, moderate, high), a decreasing trajectory, an increasing trajectory, and occasionally a curvilinear trajectory. Veldman and colleagues related INT and EXT symptom trajectories to late adolescent employment outcomes. They found that individuals with high-stable INT+EXT or EXT only trajectories (but not with other trajectories) had the highest risk of poor outcome [15], providing evidence for different associated outcomes for distinct developmental trajectories. Importantly, Veldman and colleagues demonstrated that trajectories of INT and EXT problems are both important sources of heterogeneity in functional outcomes and should ideally be accounted for at the same time. Usually, INT and EXT trajectories are examined in separate models as though they were independent outcomes. In reality, INT and EXT symptoms are correlated, individuals can differ in their relative levels of INT and EXT symptomatology, and the course of INT and EXT symptoms is associated during childhood and into adolescence [11,16–18]. The current study used Parallel-Processes Latent Class Growth Analyses (PP-LCGA) to simultaneously model the (co-)developmental trajectories of INT and EXT symptoms over the course of adolescence [19]. Two studies that previously applied this approach [17,18] found symptom codevelopment classes that were characterized by INT and EXT symptoms that did not change in tandem with each other, highlighting the importance of examining parallel processes.

**Competing interests:** The authors have declared that no competing interests exist.

## Adjusting for childhood functioning, putative confounders, and current mental health

Early-adult functioning is -at least to some extent- a continuation of childhood and adolescent functioning and this should be accounted for when examining the association of adolescent psychopathology with early–adult functioning. It is also important to control for possible confounding by relatively stable psychological, social and contextual determinants of both mental health and functioning, such as child temperament, IQ, and family environment [4,6,18,20]. The few studies with substantial long-term follow-ups that statistically adjusted for an extensive range of putative confounders typically found a reduced strength of the associations between childhood/adolescent mental health problems and (early-) adult functioning [4,6].

Likewise, it is to be expected that early-adult functioning is influenced by current mental health status, given the existing evidence on the cross-sectional association between psychopathology and functioning at any point in life [20]. Although some studies of long-term functional outcome did control for current mental health, the adjustment was typically at the diagnostic categorical level [9]. This neglects the fact that people without a (sub)clinical disorder at follow-up may still suffer from significant mental health problems but fail to meet exact diagnostic criteria. To test if the long-term effects of adolescent mental health trajectories were independent of childhood functioning, putative confounders and current mental health, we adjusted for this in our analyses.

## The current study

In sum, to better understand the links between adolescent mental health and early-adult functioning, the aim of this study was twofold. First, we aimed to identify distinct developmental trajectories of (co-occurring) INT and EXT symptoms over the course of adolescence. Second, we aimed to document the associations between these adolescent trajectories and functional outcomes in young adulthood in three domains: economic, social, and health, unadjusted and adjusted for childhood functioning, putative confounders and current mental health. To avoid operational confounding between adolescent trajectories and early-adult functional outcomes, the time frame of the trajectories was limited to adolescence (ages 11–19), whereas the functional outcomes were assessed at age 22 and typically referred to the preceding month(s).

Based on previous literature, we expected to identify a normative trajectory class with low INT and EXT symptoms, a high-severity trajectory class with high INT and EXT symptoms, and a moderate-severity class with predominantly INT symptoms. Moreover, we expected that the most severe trajectory class would be associated with the poorest outcomes and that these associations would attenuate, but remain significant, after adjustment for childhood functioning, confounding factors and current mental health.

## Methods

### Sample and procedure

The TRacking Adolescents' Individual Lives Survey (TRAILS) is a prospective cohort study of Dutch adolescents using bi- or triennial measurements from age 11 onward. Its aim is to chart and explain the development of mental health from preadolescence into adulthood. Previous publications have extensively described its design, methods, response rates and bias [21–25].

Participants were selected from five municipalities in the North of the Netherlands, both urban and rural, including the three largest cities. Children born between 1 October 1989 and 30 September 1991 were eligible for inclusion, providing they met the inclusion criteria and their schools were willing to participate [21]. Of the initially 135 primary schools in the area,

encompassing 3483 eligible children, over 90% (n = 3145 children), agreed to participate in the study. Then, parents/guardians and children were informed though information brochures about the study goals, selection procedure, confidentiality and measurements. Shortly thereafter, an interviewer contacted the parents by telephone to invite them to participate. Another 210 children were excluded because they were unable or incapable to participate (e.g., due to severe mental retardation or physical illness or handicap or if no Dutch-speaking parent was available, enrolling a total of 2935 eligible children. Through extended efforts (including telephone calls, reminders and home visits), 76% of these children and their parents consented to participate (wave 1, n = 2230, mean age = 11.1 years, SD = 0.6, 50.8% girls). Subsequent data collection waves took place bi- or triennially and had good retention rates (wave 2 mean age 13.6; 96%; wave 3 16.3, 81%; wave 4 19.1, 84%; wave 5 22.3, 80%).

**Ethical considerations.** The study was approved by the Dutch Central Committee on Research Involving Human Subjects (CCMO-NL38237.042.11; www.ccmo.nl). All children and their parents provided written informed consent to participate.

## Measures

**Adolescent mental health.** Mental health symptoms were assessed with the INT and EXT scales of the Youth Self Report (YSR) at waves 1–3 and the Adult Self Report (ASR) at wave 4 [26,27]. Participants responded to each item on a three-point scale (0 = not true, 1 = somewhat true, 2 = very often or always true). Both YSR and ASR have good validity and reliability [28,29], which were also confirmed for the Dutch versions [30]. The INT scale consists of the subscales Anxious/Depressed, Withdrawn/Depressed and Somatic Complaints (31 items for the YSR [range 0–52], 39 items for the ASR [range 0–78]). The EXT scale consists of the subscales Rule-Breaking Behaviour, Aggressive Behaviour and Intrusive (32 items for the YSR [range 0–54], 35 items for the ASR [range 0–70]). Internal consistency for the INT and EXT subscales was good to excellent across all waves (T1: $\alpha_{INT} = 0.87$, $\alpha_{EXT} = 0.85$; T2: $\alpha_{INT} = 0.88$, $\alpha_{EXT} = 0.85$; T3: $\alpha_{INT} = 0.89$, $\alpha_{EXT} = 0.87$; T4: $\alpha_{INT} = 0.93$, $\alpha_{EXT} = 0.89$.

**Early-adult outcomes.** Our aim was to identify a broad range of early-adulthood outcomes that typically impede functioning for most individuals. The choice of outcomes was primarily based on the Copeland et al. studies and our 2017 study linking childhood and adolescence psychiatric history on early-adult functioning [13]. There were some adaptations. For example, contrary to Copeland et al. [9], we did not have data on serious criminality and incarceration for all participants, but instead included substantial antisocial behavior as outcome measure. The selected outcomes covered the economic, social and health domains and were mostly generated through self-report. Table 1 describes the 17 outcomes assessed at age 22 (wave 5), on average 38 months after the wave 4 ASR (age 19y). Most outcomes refer to the 3–6 months preceding wave 5, some to the wave 4–5 interval. In some cases, the thresholds for outcomes were self-evident (e.g., early parenthood yes/no, receiving social benefits yes/no). Where such thresholds were not available (i.e., ordinal or dimensional measures), outcomes were defined to identify approximately 10% of participants or fewer (e.g., delinquency was defined as 4+ self-reported delinquent acts, problematic alcohol use was defined as a score $\geq$ 14 on the alcohol use disorder identification test) to ensure the outcome was deviant. Additional information about outcomes is available in Table 1.

**Childhood covariates.** To clarify that the effects of adolescent trajectories are independent of other childhood psychosocial factors, we accounted for the following family and individual variables:

*Childhood functioning.* For 12 out of 17 early-adult outcomes, comparable data was collected at age 11 (wave 1) as well, see for more detail about the measures that were used to

**Table 1. Definitions and prevalence of early-adult outcomes.**

| Domain | | Definition | Prevalence (%) | Age 11 childhood functioning |
|---|---|---|---|---|
| **Economic** | | | | |
| Low educational level[#] | | attained level = primary school or lower vocational/secondary | 6.8 | Special education (yes/no) |
| Frequent absenteeism[#] | | missing/skipping 7+ days in past 3 months from school/work | 6.0 | Self-rated school absence (sometimes/often) |
| Receiving social security benefits | | receiving benefits due to unemployment or disability | 7.9 | |
| Serious financial difficulties | | financial difficulties, resulting in + $\geq$ €5000 debt | 7.1 | |
| **Social** | | | | |
| Early parenthood | | becoming a parent in adolescence (age 18/19) | 6.3 | |
| Multiple social difficulties | | recent let-down experiences and interpersonal conflicts (score = 4), assessed with the TRAILS Events Checklist | 15.4 | |
| Delinquency[#] | | 4+ self-reported delinquent acts, assessed with the Antisocial Behavior Questionnaire [31] | 13.3 | Highest 10% on Antisocial Behavior Questionnaire (score $\geq$ 20, range 0–124) [31] |
| **Health** | | | | |
| Problematic substance use | Smoking[#] | 10+ cigarettes per day during past month | 14.7 | Smoked cigarettes 4+ times, assessed with the Antisocial Behavior Questionnaire [31] |
| | Alcohol[#] | highest 10% score ($\geq$14) on alcohol use disorders identification test (range 0–40) [32] | 9.1 | Used alcohol 4+ times, assessed with the Antisocial Behavior Questionnaire [31] |
| | Cannabis[#] | highest 10% score ($\geq$15) on cannabis use problems identification test (range 0–76) [33] | 10.1 | Ever used drugs (1+), assessed with the Antisocial Behavior Questionnaire [31] |
| Suicidality | | suicidal ideation or self-harm (sometimes or often) in the past 6 months, assessed with the Adult Self Report and the parent-reported Adult Behavior Checklist for ages 18–59. | 2.6 | Suicidal ideation or self-harm in the past 6 months (sometimes or often), assessed with the Youth Self Report [26] |
| Mental health concerns | Unhappiness /dissatisfaction[#] | lowest 10% score (<6) on combined happiness and satisfaction rating scales (range 2–20) | 12.1 | General unhappiness, assessed with the Self-Perception Profile for Children [36] |
| | Poor sleep quality[#] | 3+ self-reported sleep problems, assessed with the Nottingham Health Profile [34] | 8.0 | Very poor self-report sleep quality in past year |
| | Loneliness[#] | feeling often lonely in past 6 months (often), assessed with the Adult Self Report [28] | 5.7 | Often feels lonely in past 6 months, assessed with Youth Self Report [26] |
| Serious physical event | | physical assault (violence, rape; score = 2+), assessed with the TRAILS Events Checklist | 3.5 | |
| Obesity[#] | | BMI >30 [35] | 7.3 | BMI>30 |
| Poor subjective physical health[#] | | Poor/moderate self-report physical health in past month | 17.9 | Poor/moderate self-report physical health in past year |

BMI = body mass index.

[#]childhood (age 11) data is available

generate the age 11 childhood functioning variables and the thresholds that were used for binarization Table 1.

**Intelligence** was prorated using the Vocabulary and Block Design subtests of the Revised Wechsler Intelligence scale for Children (WISC-R), administered at age 11 (wave 1) [37].

**Temperament**, including Effortful Control, Frustration, and Fearfulness, was assessed at age 11 (wave 1) with the Dutch version of the parent-reported Early Adolescent Temperament Questionnaire (EATQ-R) [38,39]. Internal consistency was acceptable to good for subscales effortful control ($\alpha$ = 0.86; 11 items) and frustration ($\alpha$ = 0.74; 5 items) and questionable for fear ($\alpha$ = 0.63; 5 items).

**Low social Economic Status (SES)** of the family was measured at age 11 (wave 1) based on standardized scores for parental education level, parental profession and family income. SES was categorized as lowest 25%, middle 50%, and highest 25% [40].

**Family functioning: Family stress** has been found to adversely affect the parent–child interactions [41], which could cascade into later problems. The amount of parenting distress parents experienced was assessed at age 11 (wave 1) with the Dutch short-version (25 items) of the Parental Stress Index (PSI), the NOSI-K (Nijmegen Parental Stress Index Short Version) [42]. This version of the PSI contains 25 items that could be rated on a 6-point scale ranging from 1 = disagree very much to 6 = agree very much. The subscale PSI Parent, consisting of 10 items referring to parent characteristics within the caregiving context, was included in our analyses. Internal consistency was good ($\alpha$ = 0.86). Relations exist between low levels of parental warmth/involvement and high levels of child externalizing and internalizing symptoms [43].The child's perception of his/her upbringing, **including parental warmth and rejection**, was assessed at age 11 (wave 1) with the Egna Minnen Beträffande Uppfostran (Swedish for 'My Memories of Upbringing') for Children *(EMBU-C)*(35)*)* [44]. The questionnaire contains 47 items asked separately for both father and mother. Given that the answers for both parents were highly correlated ($r$s = 0.69 for Rejection and 0.79 for Emotional Warmth), we used the combined total score in our analyses. Internal consistency was good to excellent ($\alpha_{warmth}$ = 0.91; 18 items, and $\alpha_{rejection}$ = 0.84; 17 items).

**Early-adult covariates.** Current mental health at the time of the assessment of early adulthood functioning, was assessed with the parent-reported Adult Behaviour Checklist (ABCL) [27,28]. We used a different informant (parent) for current mental health to avoid method variance with the self-reported early-adult outcomes. Because some of the items of the subscales overlapped with some of outcome variables (i.e., the items 'Sleep problems', 'Talks (or thinks) about killing self', 'Unhappy, sad or depressed', 'Fights a lot', 'Physically attacks people', 'Uses drugs for non-medical purposes', 'Drinks too much alcohol or gets drunk', 'Fails to pay debts'), we excluded those items from the wave 5 ABCL INT and EXT scales. Cronbach's alpha of the adjusted ABCL-INT and -EXT scales were excellent (both: $\alpha$ = 0.91).

**Descriptive variables.** *Lifetime mental health care (MHC) use*. Lifetime MHC use was based on parent-report, collected at wave 1 (concerning MHC use prior to wave 1) through wave 4 (each time concerning the time interval between adjacent assessment waves).

*Diagnostic information*. The presence of lifetime mental disorders was assessed at age 19 (wave 4) using the Composite International Diagnostic Interview (CIDI) 3.0. The CIDI is a structured diagnostic interview that has been used in multiple surveys worldwide to generate diagnoses based on the DSM-IV [45]. The assessment included mood disorders, anxiety disorders, behaviour disorders, and substance dependence. The interview was administered by trained lay-interviewers. Of the current sample (n = 1524), 91.3% provided CIDI data.

## Statistical analysis

First, PP-LCGA was run on the wave 1–4 INT and EXT scale scores simultaneously to identify distinct developmental INT and EXT trajectory classes. Not all wave 1–4 YSR/ASR INT and EXT scales were normally distributed, therefore, a Van der Waerden transformation was applied to normalize these measures. A model-configuration was used with one latent-class variable influencing both the INT and EXT trajectory parameters (i.e. mean intercepts and slopes). Quadratic terms were also included in all models to enable the modelling of non-linear growth. Models with increasing numbers of classes were estimated and model-fit and usefulness were compared using the Akaike Information Criterion (AIC), the Bayesian Information Criterion (BIC), classification quality (entropy), and the Bootstrapped Likelihood Ratio Test (BLRT) [46]. Finally, the usefulness of each class-solution was evaluated by checking the extent of qualitative differentiation between classes and the number of patients in the smallest class. Models were estimated using the Full Information Maximum Likelihood Estimation (FIML)

estimator with robust standard errors (MLR) [19,47]. Models were run with multiple sets of random starting values to prevent identification of models at local maxima. All PP-LCGAs were run with Mplus, Version 6 [19].

The PP-LCGA analyses were based on 1,524 individuals who provided YSR/ASR INT and EXT data and data on at least five early-adult outcomes. The classes of the best fitting model were labelled according to their INT and EXT symptom trajectory characteristics. The classes were pairwise compared on descriptives and covariates, and on the percentage of individuals reported any and multiple (2+) poor outcomes. Second, the associations of trajectory membership with early-adult outcomes were investigated, using binary logistic regression analyses, using the trajectory classes as predictor variables. We created dummy variables, each representing a trajectory class, with the trajectory that included the lowest levels of both INT and EXT symptoms (i.e., the 'healthiest' trajectory) as the reference group. Dependent variables were the early-adult outcome variables as described in Table 1. We ran a separate analysis for each outcome. In the unadjusted model, the dependent variables were, separately, regressed on the trajectory dummy variables. The adjusted model included the following covariates: childhood functioning, child covariates, and current INT and EXT symptoms. In the adjusted model, the dependent variables were, separately, regressed on the trajectory dummies and all covariates simultaneously. We used the standardized residual variance of current mental health that we obtained by regressing trajectory class membership on the parent- reported INT and EXT scales in a separate analysis and saving the standardized residuals. This way, we adjusted for the correlation between adolescent trajectories and current mental health. The correlation between residual parent-reported INT and EXT scales and trajectory class membership was small (r = .10, and r = .08, respectively, both *p*-values < .001). We conducted all regression analyses with IBM SPSS, version 23.

## Results

### Missing data

Missing self-reported INT and EXT data was <5% for wave 1, 2, 4 and 5 and 10% for wave 3. About 10% of parent-reported current INT and EXT was missing. However, missing data was handled in Mplus using FIML, thus all 1,524 individuals who were included in the PP-LCGA analyses, were assigned to a trajectory class. Data was missing for <2% of early-adult outcomes, <3% of childhood functioning <2%, for IQ, family SES, and perceived parenting, 5.6% for parenting stress, 8.5% for temperament, and 8.7% for lifetime mental health disorder (CIDI) data. Missing outcome, childhood functioning, confounder, and current mental health data was not imputed, and all regression analyses were performed on complete cases.

### Model comparisons

Table 2 provides the fit indices for the PP-LCGA class-solutions. In the PP-LCGA, the AIC and BIC decreased and the BLRT remained significant (*p* < .0001) with each class addition, giving no definitive clues as to which model was most optimal. Therefore, model interpretability of the models was considered to select the most optimal solution. The 4-, 5- and 6-class models showed at least two codevelopment classes that were characterized by INT and EXT symptoms that did not change in tandem with each other, making them conceptually more interesting than the 2-, and 3-class models. The 7- and 8-class models were not further considered due to small class-sizes (at least one class with n<10%) and estimation problems.

The 4-class model showed a high-severity trajectory class with individuals displaying continuously high INT and EXT symptoms throughout adolescence ('Continuous moderate-high INT + EXT'; n = 298), a low-severity trajectory class with individuals with decreasing low INT

**Table 2. Fit indices for the PP-LCGA based on YSR and ASR wave 1–4 INT and EXT subscales.**

| #c | Fit indices | | | | |
|----|------|------|------|---------|----------------|
|    | AIC | BIC | BLRT | entropy | Smallest class |
| 1 | 32373.621 | 32448.228 | - | - | 1524 |
| 2 | 30037.283 | 30149.194 | < .0001 | .789 | 690 |
| 3 | 29559.321 | 29708.536 | < .0001 | .756 | 242 |
| 4 | 29289.988 | 29476.506 | < .0001 | .712 | 298 |
| 5 | 29091.051 | 29314.873 | < .0001 | .708 | 187 |
| 6 | 28903.303 | 29164.429 | < .0001 | .709 | 178 |
| 7 | 28756.655 | 29055.084 | < .0001 | .716 | 106 |
| 8 | 28666.440 | 29002.173 | < .0001 | .712 | 92 |

AIC = Akaike Information Criterion, BIC = Bayesian Information Criterion., BLRT = Bootstrapped Likelihood Ratio Test. Based on fit indices, combined with the most informative trajectory profiles, the 4-class model was chosen for further analyses

and EXT symptoms throughout adolescence ('decreasing-low INT + EXT'; n = 460; the health-iest and thus reference class, 30% of the sample), a moderate-severity trajectory class with more severe INT than EXT symptoms over time ('continuous moderate, INT>EXT'; n = 414), and a decreasing-severity trajectory class with more severe EXT than INT symptoms over time ('decreasing moderate, EXT>INT'; n = 352). The 5-class model showed similar classes but added an in-between group with a similar decreasing-severity more EXT than INT trajectory, but with low to moderate overall levels of EXT and INT symptoms. The majority of individuals from the decreasing low (reference) class in the 4-class model were transferred to this new in-between class, leaving a reference class of n = 187 (12% of total sample). The 6-class model showed similar trajectory classes as the 5-class model, but added an additional increasing-severity trajectory class with more severe EXT than INT. Also in this model, the healthiest class comprised 12% of the sample (n = 189). The 5-class model was not further considered since no qualitatively distinct class emerged in this model compared to 4-class model. This was the case for the 6-class model, leaving the 4- and 6-class models as viable options. We debated about choosing the more parsimonious 4-class model, with the risk of being overly reduction-istic in reducing the variability in adolescent development to four classes. Or choosing the more fine-grained 6-class model, with the risk of estimations problems due to the much smaller reference group combined with low exposure rates for some outcomes, and in addi-tion, multiple comparisons between quantitatively but not qualitatively distinct classes that are unlikely to reveal differential effects on functioning. In the end, the more parsimonious 4-class model was chosen over the 6-class model for further analyses, based on the substantive mean-ingfulness and interpretability of the classes. However, because the additional increasing, EXT>INT class from the 6-class model has the potential of differential associations with func-tional outcomes compared to the other, qualitatively distinct, trajectories from the 4-class model, we decided to run the analyses on the 6-classes and document their associated func-tional outcomes (findings of the other trajectories are not presented in the main text, but can be found in the Supplemental material S1 Fig and S1 Table).

To facilitate clinical interpretation of the distinct INT and EXT trajectories, we linked the classes' YSR/ASR mean scores back to T-scores. T-scores above 60 and 64 respectively are con-sidered sub-clinically and clinically meaningful. Fig 1 shows the LCGA trajectories (based on mean T-scores) for each of the PP-LCGA classes on each of the two investigated symptoms domains (INT and EXT). It shows that the means for the 'continuous moderate-high INT+ EXT' class fell within (internalizing) or were close to (externalizing) the subclinical cut-off level (i.e., T = 60). For the other three classes, mean T-scores fell within the normal range.

## Internalizing symptoms

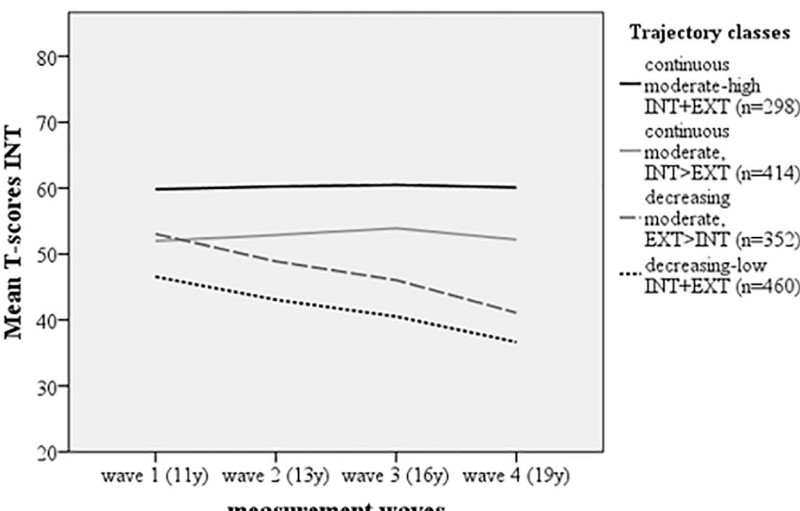

## Externalizing symptoms

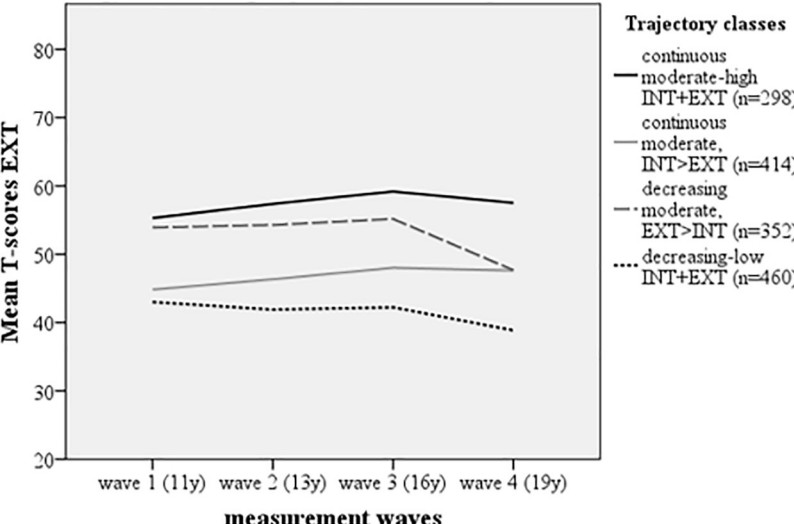

**Fig 1. Parallel-processes Latent Class Growth trajectories of simultaneously modelled internalizing (INT) and externalizing (EXT) symptoms, based on mean T-scores.** Subclinical and clinical cut-off scores for the YSR/ASR are T = 60 and T≥64 respectively.

### Quantitative differences between trajectory classes

Compared to the 'decreasing-low INT+EXT' class, the other three classes were similarly characterized by lower effortful control and higher frustration and fear, higher parenting stress, less parental warmth and more parental rejection, and higher percentages of mental health care use and lifetime INT and EXT disorders with the 'continuous moderate-high INT+EXT' class being the most problematic followed by the 'decreasing moderate, EXT>INT' class. The 'continuous moderate-high INT + EXT' and the 'continuous moderate INT>EXT' classes

Table 3. Characteristics of the adolescent mental health trajectory classes.

| | Trajectories | | | |
|---|---|---|---|---|
| | 1. Decreasing-low INT+EXT (n = 460) | 2. Decreasing moderate, EXT>INT (n = 352) | 3. Continuous moderate INT>EXT (n = 414) | 4. Continuous moderate-high INT+EXT (n = 298) |
| Sex (% males) | 51.1% | 61.1% | 27.1% | 34.9% |
| Temperament | | | | |
| Effortful control | 3.45 (0.67) | 3.19 (0.64) | 3.30 (0.65) | 3.12 (0.71) |
| Frustration | 2.60 (0.62) | 2.86 (0.61) | 2.75 (0.65) | 2.95 (0.66) |
| Fear | 2.29 (0.71) | 2.35 (0.67) | 2.44 (0.73) | 2.61 (0.73) |
| IQ | 100.56 (14.12) | 100.42 (14.39) | 98.57 (14.96) | 99.16 (15.16) |
| Low family SES (%) | 19.2% | 19.0% | 20.3% | 19.2% |
| Parenting stress | 1.59 (0.66) | 1.83 (0.78) | 1.68 (0.71) | 2.08 (0.90) |
| Parental warmth | 3.30 (0.47) | 3.18 (0.50) | 3.30 (0.49) | 3.24 (0.49) |
| Parental rejection | 1.39 (0.24) | 1.53 (0.33) | 1.42 (0.24) | 1.63 (0.34) |
| Lifetime MHC use (%) | 23.8% | 37.5% | 39.1% | 61.1% |
| % INT disorders [CIDI] | 16.5% | 23.0% | 46.7% | 66.7% |
| % mood disorders | 3.6% | 9.9% | 19.6% | 42.6% |
| % anxiety disorders | 13.4% | 18.0% | 37.9% | 48.5% |
| % EXT disorders [CIDI] | 3.8% | 23.3% | 9.9% | 40.4% |
| % behavioural disorders | 3.8% | 18.6% | 8.9% | 31.1% |
| % substance dependence | 0.2% | 9.0% | 2.1% | 18.5% |

EXT = externalizing behaviour, INT = internalizing behaviour, IQ = intelligence quotient; SES = socio economic status, MHC = mental healthcare, CIDI = Composite International Diagnostic Interview [45]

consisted of more females, whereas the 'decreasing moderate, EXT>INT' class had a higher percentage of males, see Table 3.

## Predicting early-adult functioning by adolescent trajectories and current mental health

Fig 2 presents the distribution of any and multiple poor outcomes across the four trajectory classes. Finding showed that of the individuals with decreasing-low INT and EXT symptoms throughout adolescence, 45.0% reported a poor adult outcome and 20% reported two or more poor outcomes. In comparison, of the individuals with continuously moderate-high levels of both INT and EXT symptoms, 84.6% reported a poor outcome and 62.8% reported two or more. Of the two other classes, around 70% of the individuals reported one poor outcome and around 40% two or more poor outcomes.

Table 4 presents the Odds Ratio's (ORs) for the logistic regression analyses on the outcome variables and summary indices. Compared to the reference group, the other three developmental trajectories predicted an accumulation of relatively poor early-adult outcomes with the strongest effects observed for individuals with continuous moderate-high levels of both INT and EXT symptoms throughout adolescence. Individuals with the 'continuous moderate, INT>EXT' trajectory were particularly at increased risk of poor mental health and to a lesser extent at risk of a poor social outcome. Individuals with the 'decreasing moderate, EXT>INT' trajectory were particularly at increased risk of poor economic outcomes and problematic substance use.

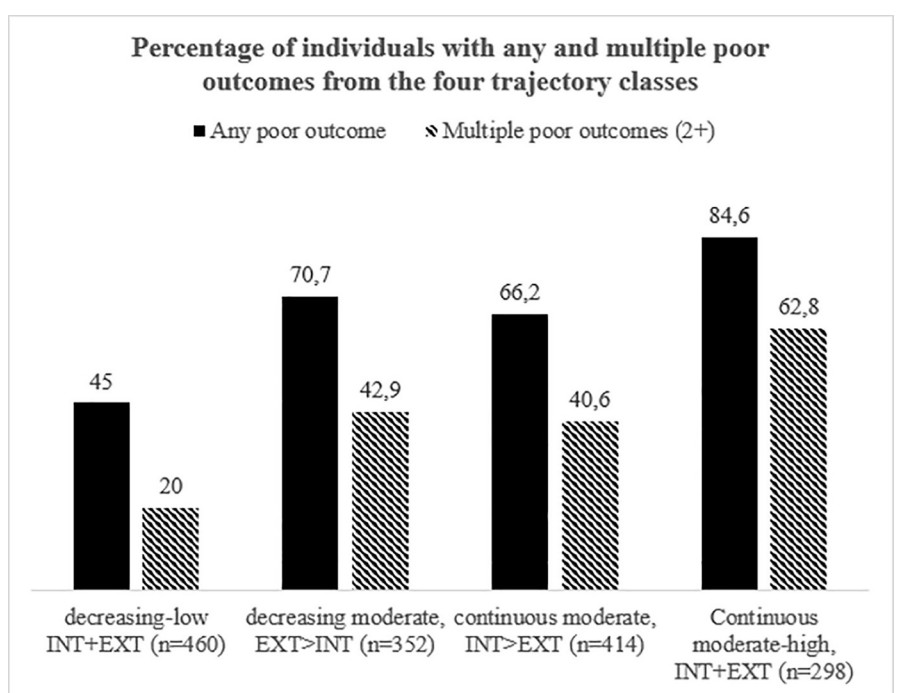

**Fig 2. Association between early-adult outcomes and adolescent mental health trajectory.**

Adjustment for childhood functioning, confounders and current mental health did not substantially alter the results. Most ORs hardly dropped in size, although current mental health explained additional variance in early-adult outcomes over and above the variance accounted for by the trajectories, see Table 4. Both current INT and EXT predicted an accumulation of economic outcomes, whereas poor mental health outcomes were only predicted by current INT, and poor social and problematic substance use outcomes were only predicted by current EXT.

The additional increasing EXT>INT class from the 6-class model was at an increased risk of poor economic outcomes, social difficulties, delinquency, problematic substance use, mental health concerns and poor subjective health.

## Discussion and conclusion

The aims of this prospective study were (1) to identify distinct developmental trajectories of (co-occurring) internalizing and externalizing mental health symptoms over the course of adolescence (ages 11–19), and (2) to document the associations between these adolescent trajectories and economic, social, and health outcomes in young adulthood (age 22), unadjusted and adjusted for childhood functioning, putative confounders and current mental health. In agreement with our hypothesis, we identified a normative trajectory class with decreasing-low INT and EXT symptoms, a trajectory class with moderate-high INT and EXT symptoms, and a moderate-severity class with predominantly INT symptoms. In addition, we identified a trajectory class with predominantly EXT symptoms that decreased in severity across adolescence. Similar to [17,18], we found two trajectory classes with INT and EXT symptoms that did not change in tandem with each other, highlighting the importance of examining parallel processes. Also, in line with our hypothesis, was that compared to the decreasing-low trajectory, the other three trajectories generally predicted less optimal early-adult outcomes, with the strongest effects observed for individuals with continuous moderate-high levels of both INT

**Table 4. Unadjusted and adjusted associations between the adolescent mental health trajectory classes and early-adult functioning outcomes.** The decreasing-low INT+EXT class (n = 460) was used as a reference.

| Domain | | Continuous moderate-high INT+EXT (n = 298) vs. reference | | Continuous moderate INT>EXT (n = 414) vs. reference | | Decreasing moderate, EXT>INT (n = 352) vs. reference | | Explained variance (R²) | | Estimates (OR; 95% CI) for current INT and EXT symptoms (covariates in adjusted analyses) |
|---|---|---|---|---|---|---|---|---|---|---|
| | | Unadj. | Adj. | Unadj. | Adj. | Unadj. | Adj. | Unadj. | Adj. | Adj. |
| | | OR (95% CI) | OR (95% CI) | OR (95% CI) | OR (95% CI) | OR (95% CI) | OR (95% CI) | | | |
| **Economic** | | | | | | | | | | |
| Low educational level | | 4.34 (1.97–9.60) | 4.33 (1.75–10.70) | 2.80 (1.27–5.81) | 2.65 (1.10–6.39) | 4.13 (1.90–8.99) | 4.45 (1.91–10.39) | .04 | .31 | INT: 6.41 (2.85–22.18) EXT: 2.14 (0.57–7.99) |
| Frequent absenteeism | | 2.57 (1.31–5.06) | 2.00 (0.94–4.27) | 1.66 (0.84–3.27) | 1.54 (0.76–3.13) | 1.40 (0.67–2.91) | 1.50 (0.69–3.25) | .02 | .08 | INT: 3.04 (0.94–9.83) EXT: 2.29 (0.59–8.82) |
| Receiving social security benefits[a] | | 3.59 (1.76–7.31) | 2.30 (0.99–5.31) | 2.41 (1.19–4.90) | 2.59 (1.17–5.73) | 1.90 (0.88–4.08) | 1.44 (0.61–3.41) | .03 | .28 | INT: 8.15 (2.68–24.81) EXT:5.80 (1.70–19.71) |
| Serious financial difficulties[a] | | 11.70 (4.48–30.56) | 11.16 (4.06–30.67) | 5.39 (2.02–14.39) | 5.94 (2.17–16.30) | 5.26 (1.93–14.34) | 5.19 (1.86–14.48) | .08 | .15 | INT: 1.80 (0.58–5.57) EXT: 6.15 (1.84–20.59) |
| **Social** | | | | | | | | | | |
| Early parenthood[a] | | 3.90 (1.88–8.12) | 3.04 (1.35–6.85) | 1.49 (0.67–3.33) | 1.18 (0.51–2.71) | 1.94 (0.88–4.29) | 2.26 (0.98–5.23) | .04 | .16 | INT: 0.78 (0.20–3.05) EXT: 4.78 (1.12–20.34) |
| Multiple social difficulties[a] | | 5.81 (3.49–9.68) | 5.60 (3.19–9.84) | 3.18 (1.91–5.29) | 3.26 (1.91–5.56) | 2.95 (1.74–5.01) | 3.41 (1.95–5.97) | .07 | .17 | INT: 1.09 (0.46–2.58) EXT: 11.55 (4.45–29.95) |
| Delinquency | | 1.85 (1.16–2.95) | 2.08 (1.20–3.62) | 0.77 (0.46–1.27) | 1.27 (0.74–2.18) | 1.70 (1.08–2.66) | 1.29 (0.78–2.12) | .03 | .16 | *INT: 0.20 (0.06–0.62)* EXT: 23.07 (8.02–66.37) |
| **Health** | | | | | | | | | | |
| Problematic substance use | Smoking | 3.83 (2.29–6.39) | 3.81 (2.17–6.68) | 1.48 (0.85–2.56) | 1.49 (0.85–2.64) | 3.99 (2.43–6.54) | 3.89 (2.31–6.54) | .07 | .13 | *INT: 0.28 (0.10–0.82)* EXT: 5.07 (1.85–13.89) |
| | Alcohol | 1.34 (0.76–2.38) | 1.34 (0.70–2.59) | 0.78 (0.43–1.40) | 1.18 (0.63–2.23) | 2.18 (1.32–3.60) | 1.54 (0.89–2.67) | .03 | .18 | *INT: 0.23 (0.06–0.92)* EXT: 3.73 (1.15–12.14) |
| | Cannabis | 3.21 (1.85–5.58) | 5.67 (2.99–10.74) | 0.88 (0.46–1.69) | 1.62 (0.81–3.24) | 2.99 (1.74–5.13) | 3.00 (1.68–5.35) | .06 | .20 | INT: 1.06 (0.35–3.21) EXT: 6.31 (2.18–18.31) |
| Suicidality | | 11.30 (3.31–38.62) | 10.66 (2.83–40.22) | 3.94 (1.08–14.44) | 4.29 (1.08–16.96) | 0.46 (0.05–4.46) | 0.43 (0.04–4.27) | .12 | .26 | INT: 8.40 (2.04–35.55) EXT: 2.99 (0.54–16.49) |
| Mental health concerns[b] | Unhappy/ dissatis-faction | 4.68 (2.81–8.09) | 5.12 (2.82–9.32) | 2.48 (1.45–4.23) | 2.80 (1.58–4.97) | 1.34 (0.72–2.48) | 1.47 (0.77–2.80) | .07 | .17 | INT: 14.77 (6.01–36.30) EXT: 1.18 (0.41–3.39) |
| | Poor sleep quality | 5.12 (2.44–10.76) | 4.71 (2.09–10.59) | 4.92 (2.42–10.03) | 5.10 (2.42–10.72) | 1.83 (0.79–4.23) | 1.93 (0.81–4.59) | .07 | .16 | INT: 6.29 (2.30–17.23) EXT: 2.64 (0.79–8.79) |
| | Loneliness | 27.50 (6.50–116.41) | 37.98 (8.54–168.90) | 21.67 (5.17–90.93) | 30.61 (7.07–132.42) | 2.79 (0.51–15.34) | 3.52 (0.63–19.70) | .16 | .26 | INT: 15.31 (5.09–46.04) EXT: 1.89 (0.48–7.40) |
| Serious physical event[a] | | 4.72 (1.49–14.99) | 4.39 (1.27–15.13) | 3.54 (1.13–11.09) | 3.44 (1.07–11.05) | 4.29 (1.37–13.44) | 4.44 (1.37–14.44) | .04 | .10 | INT: 1.10 (0.21–5.69) EXT: 4.28 (0.78–23.39) |

*(Continued)*

**Table 4.** (*Continued*)

| | | Continuous moderate-high INT+EXT (n = 298) vs. reference | | Continuous moderate INT>EXT (n = 414) vs. reference | | Decreasing moderate, EXT>INT (n = 352) vs. reference | | Explained variance ($R^2$) | | Estimates (OR; 95% CI) for current INT and EXT symptoms (covariates in adjusted analyses) |
|---|---|---|---|---|---|---|---|---|---|---|
| | | Unadj. | Adj. | Unadj. | Adj. | Unadj. | Adj. | Unadj. | Adj. | Adj. |
| Obesity | | 2.25 (1.26–4.04) | 1.33 (0.67–2.63) | 1.00 (0.53–1.88) | 0.71 (0.36–1.41) | 1.19 (0.63–2.24) | 1.03 (0.51–2.08) | .02 | .18 | INT: 1.02 (0.30–3.50) EXT: 3.14 (0.85–11.64) |
| Poor subjective physical health | | 5.71 (3.53–9.25) | 5.62 (3.32–9.52) | 4.30 (2.71–6.84) | 4.13 (2.55–6.70) | 2.01 (1.19–3.39) | 2.29 (1.33–3.94) | .10 | .16 | INT: 5.56 (2.49–12.38) EXT: 0.99 (0.38–2.56) |

OR = odds ratio, 95% CI = 95% confidence interval, INT = internalizing, EXT = externalizing, Unadj. = unadjusted model, Adj. = adjusted model (covariates were: childhood functioning, sex, IQ, temperament (effortful control, fear, frustration), family SES, parenting stress, perceived parenting (warmth and rejection), and current mental health).

Covariates printed in *italic* had a protective effect as indexed by an OR<1

[a] adjusted analyses controlled for all covariates except childhood functioning data

[b] adjusted analyses controlled for all covariates except current mental health

and EXT symptoms throughout adolescence. The effects of the other two trajectories varied with outcome.

The associations between adolescent mental health and early-adult outcome hardly attenuated after controlling for pre-adolescent functioning and selected confounders. This strongly suggest that the observed associations between mental health trajectories during adolescence and early-adult functioning are not explained by other known correlates of functioning, such as childhood functioning and correlates of both psychopathology and functioning, such as sex, IQ, temperament, family SES, parenting stress, and parenting style. Finally, current mental health status accounted for additional variance in early-adult outcomes, but this did not overlap strongly with the trajectories of mental health in adolescence, which remained important as well. This suggests that both past and current mental health are important for understanding functioning in early adolescence.

## Implications

Taken together, our results suggest that individuals with high levels of INT and EXT symptoms throughout adolescence are at increased risk of a suboptimal transition into adulthood, as indexed by higher odds for relatively poor early-adult outcomes. Close to two thirds of these individuals reported two or more adverse outcomes, compared to one in five out of individuals with low levels of INT and EXT symptoms. It is known from the literature that individuals with a history of childhood and adolescent psychiatric problems continue to display impairment into adulthood [5].Not surprisingly, the 'continuous moderate-high INT+EXT' class also had the highest percentages of mental health diagnoses. Our findings further suggest that adverse long-term outcomes were not limited to individuals with continuously high problem levels during adolescence. Also individuals whose symptoms significantly improved (decreased) with age (i.e., decreasing-moderate, EXT>INT class) and whose predominantly INT symptoms were moderate-severe (i.e., continuous moderate, INT>EXT class) were at increased risk, although less so. From these two trajectory classes, around 40% reported two or more poor outcomes. This implies that there are long-term effects of childhood and adolescent mental problems on early-adult functioning, even if the problems do not (or to a lesser extent) persist into adulthood, even if the symptoms are subthreshold, and even after statistically

controlling for pre-adolescent functioning, confounders and current mental health symptoms, corroborating the earlier findings of Copeland and colleagues [9]. This highlights the need to reduce childhood vulnerabilities and shore up resilience to increase opportunity and optimal outcomes. Interventions to prevent further escalation of problem behavior need to begin early (ideally prior to the development of significant mental health problems) [48].

That being said, there was no one-to-one relationship between adolescent mental health and early-adult functional outcomes. Not all individuals with elevated mental problem levels report poor outcomes. Whereas youth with persistently high problem levels had a three-fold increased risk of multiple poor outcomes compared to youth with persistently low problem levels (62.8% versus 20%), over one third of them had fewer than two poor outcomes. This indicates substantial functional resilience, even in this group with high levels of self-reported mental problems. This suggests that experiencing psychopathology in adolescence does not necessarily result in a compromised transition into adulthood for everyone. Why childhood and/or adolescent problems have negative long-term consequences in some individuals and not others is an important question for further research.

This additional increasing EXT>INT class was at similar increased risk of the continuous moderate, INT+EXT class; despite lower INT levels and much lower EXT symptom levels in early adolescence, the associated outcomes hardly differed. This suggest that predominantly the presence of EXT symptoms, particularly during late adolescence, may drive poor early-adult functioning. However, these findings should be interpreted with caution given that many estimates had rather large confidence intervals, possibly due to the much smaller reference sample for the increasing EXT>INT class.

## Which type of problems drives which outcomes

The poorest outcomes were found for individuals with continuously high levels of both INT and EXT symptoms through adolescence, across the three outcome domains. Out of the 17 outcomes, 16 were affected with 13 ORs exceeding 3.0. Somewhat better outcomes occurred in individuals with continuous moderate levels of INT symptoms compared to EXT symptoms and in individuals with (decreasing moderate) higher levels of EXT than INT symptoms throughout adolescence, although respectively still ten and nine (seven after adjustment for covariates) outcomes were affected. We found some differential associations with outcomes for these two trajectory classes. Both were associated with poor economic outcomes, social difficulties, physical events and poor subjective health. However, while individuals with elevated levels of externalizing symptoms during adolescence were at increased risk of problematic substance use, individuals with increased levels of internalizing symptoms during adolescence were at increased risk of mental health concerns (unhappiness, poor sleep quality and loneliness). These findings follow a pattern of associations that is largely consistent with existing literature on long-term consequences of internalizing problems such as depression [49], and externalizing problems such as conduct disorder [50].

Current mental health accounted for additional variance in early-adult outcomes over and above the variance accounted for by the adolescent trajectories. A similar pattern was found for current mental health symptoms with internalizing problems predicting in particular mental health concerns, while externalizing problems were more relevant for social difficulties and problematic substance use.

## Strengths and limitations

The study had many strengths including the well-documented sample of adolescents followed from preadolescence to early adulthood, the simultaneous consideration of both INT and EXT

problems, and the breadth and heterogeneity of early-adult outcomes. However, the study also has limitations: (1) there were no indicators of childhood functioning for five early-adult outcomes, although some were not applicable at that age (e.g., receiving social benefits). (2) Trajectory information was only available from age 11 onwards. (3) Despite moderate non-response at baseline and limited attrition at follow-ups, both were not random but predicted by male gender, non-Western ethnicity, low SES, low IQ and academic achievement, poor physical health, and behaviour and substance use problems [22]. (4) Parents may not be the best informant of the mental health status of their 22-year old children, because the child may no longer live at home or have chosen not to disclose mental health issues to their parent(s). Low to moderate parent-child agreement is found in most studies describing discrepancies in reports of mental health disorders [51]. We chose parent-reported current mental health to avoid shared method variance with the self-reported early-adult outcomes. In our sample, the correlation between self- and parent-reported current mental health problems was $r \approx .40$, which is similar or larger than that typically found at younger ages. (5) Some of the estimates had rather large upper confidence limits, likely due to the low exposure rates of these outcomes within the classes, suggesting that these findings should be interpreted with caution. (6) The trajectory classes form a simplified description of the true trajectory variations in the sample. LCGA assumes local independence, meaning that all interpersonal trajectory variations are assumed to be solely explained by latent class membership. Interpersonal variations are likely to be dimensional and more complex. (7) Finally, our findings do not provide causal evidence that mental health problems during adolescence influenced early-adult functioning. We cannot exclude the possibility that the higher rate of poor outcomes in affected youth is a consequence of unmeasured risk factors or the result of prior events and experiences that have influenced both (the developmental of) adolescent psychopathology and functional outcomes.

## Recommendations for future research

Adding to existing literature on long-term associations between adolescent mental health problems and poor adult functional outcomes, our study has shown that these associations are not limited to individuals with continuously high levels of mental health problems, but are also present in individuals with moderate symptom trajectories, not attenuated by current mental health symptoms, and that some of the outcomes were differentially associated with distinct symptom trajectories (e.g., problematic substance use problems were only significantly associated with trajectories including elevated EXT symptoms). What we did not address in this study are the potential mechanisms driving the association between adolescent mental health issues and early-adult functional outcomes (that are also likely to be interrelated). For example, IQ may have an impact on externalizing trajectories which may then have an impact on substance use and subsequently unemployment. Examining these underlying pathways would give important insights into the factors that drive the associations and point towards potential targets for prevention and intervention efforts, but it was beyond the scope of the current study.

## Conclusions

This study sought to explain early-adult functioning from information on the trajectory of mental health problems during adolescence and current mental health problems. Findings showed that both current and past mental health problems were strongly related to adult functioning even if these symptoms decreased over adolescence, and even if symptom levels did not meet diagnostic criteria, extending the findings of Copeland and colleagues [9]. Our findings document that even adjustment for current mental health problems, irrespective whether

these meet diagnostic criteria, does not remove the impact of adolescent mental health problems on early-adult functioning. Pooled together, our findings indicate that current and adolescent externalizing problems impact most on social difficulties and problematic substance use while current and adolescent internalizing symptoms hamper mostly mental health outcomes. Economic outcomes are affected by both. A little over 7% of our youth had five or more poor early-adult outcomes and is probably at risk of an unsuccessful transition into adulthood. This suggest a need to reduce vulnerabilities and shore up resilience. Important targets may be traits related to self-control and neuroticism, which seem modifiable to some extent [52–55]. Improved mental health during adolescence may help to cope with adult developmental challenges.

## Supporting information

**S1 Table. Unadjusted and adjusted associations between the adolescent mental health trajectory classes (derived from the 6-class model) and early-adult functioning outcomes.** The decreasing-low INT+EXT class (n = 189) was used as a reference. OR = odds ratio, 95% CI = 95% confidence interval, INT = internalizing, EXT = externalizing, Unadj. = unadjusted model, Adj. = adjusted model (covariates were: childhood functioning, sex, IQ, temperament (effortful control, fear, frustration), family SES, parenting stress, perceived parenting (warmth and rejection), and current mental health). Covariates printed in *italic* had a protective effect as indexed by an OR<1. [a]adjusted analyses controlled for all covariates except childhood functioning data. [b]adjusted analyses controlled for all covariates except current mental health. (PDF)

**S1 Fig. Parallel-processes Latent Class Growth trajectories of simultaneously modelled internalizing (INT) and externalizing (EXT) symptoms from the 6-class model, based mean T-scores.** Subclinical and clinical cut-off scores for the YSR/ASR are T = 60 and T≥64 respectively. (TIF)

## Acknowledgments

This research is part of the TRacking Adolescents' Individual Lives Survey (TRAILS). Participating centers of TRAILS include various departments of the University Medical Center and University of Groningen, the University of Utrecht, the Radboud Medical Center Nijmegen, and the Parnassia Group, all in the Netherlands. TRAILS has been financially supported by various grants from the Netherlands Organization for Scientific Research (NWO), ZonMW, GB-MaGW, the Dutch Ministry of Justice, the European Science Foundation, the European Research Council, BBMRI-NL, and the participating universities. We are grateful to everyone who participated in this research or worked on this project to make it possible.

## Author Contributions

**Conceptualization:** Anoek M. Oerlemans, Johan Ormel.

**Data curation:** Dennis Raven.

**Formal analysis:** Anoek M. Oerlemans, Klaas J. Wardenaar.

**Methodology:** Klaas J. Wardenaar, Catharina A. Hartman.

**Supervision:** Johan Ormel.

**Visualization:** Anoek M. Oerlemans.

**Writing – original draft:** Anoek M. Oerlemans, Johan Ormel.

**Writing – review & editing:** Anoek M. Oerlemans, Klaas J. Wardenaar, Dennis Raven, Catharina A. Hartman, Johan Ormel.

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
