## [Decision Letter · Decision Letter 0]

18 Nov 2019

PONE-D-19-24355

The association of developmental trajectories of adolescent mental health with early-adult functioning

PLOS ONE

Dear Dr Oerlemans,

Thank you for submitting your manuscript to PLOS ONE. After careful consideration, we feel that it has merit but does not fully meet PLOS ONE’s publication criteria as it currently stands. Therefore, we invite you to submit a revised version of the manuscript that addresses the points raised during the review process.

Points required to address in the revised manuscript

**Reviewer 1**

General major comments

1. There is no consistency regarding page numbers throughout the paper, e.g. there are several pages 1 and 2 and at some time the page number jumps from 2 to 34, which makes it difficult to provide specific amendments.

2. The main Aim of the paper is not clear and consistently described throughout the paper. In the abstract the aim is describes as to investigate if there is an association between adolescent mental health trajectories and early-adult functioning. In the introduction the aim focuses on the attenuating separate effects of three domains of confounders. Third, in the beginning of the Discussion section the following is written: "In agreement with out hypothesis, we identified a normative trajectory class with decreasing-low INT and EXT symptoms…..", but no specific hypothesis regarding the mental health trajectories has been formulated?.

3. The arguments for choosing the 17 outcomes are very limited. In p.7 L 166 is written: ' A broad range of early-adulthood outcomes was identified'. A much more detailed description of the term 'functional outcome' is needed, as well as arguments for the choice of outcomes based on a literature review and an explanation of how the individual outcomes are generated and on which basis the thresholds were chosen.

4. The degree of description of the different variables varies a lot. Some of the exposure variables are described in details, whereas the outcome variables are only briefly described.

Major comments

Abstract

The background only contains the aim of the study. A few sentences of background should be added, stating the importance of this study. More compelling is the fact that the aim is not identical with the aim described in the end of the Introduction section or the Discussion section, as mentioned above.

Introduction

In the end of the first paragraph of the Introduction INT and EXT problems are mentioned as part of the aim, before the concepts are explained. This section could be deleted since the aim is already stated in the last paragraph of the Introduction.

Methods

The Sample and procedure section could use a few lines describing the sampling procedures of the Trails. How participants were selected and contacted (e-mail, post?)

p.6, L 128 & : describe the construction of the INT and EXT scales and outcomes in more details. Number of items? Range? scores?

Results

I would recommend not testing for or indicating statistical significance by *. It is only relevant to present and discuss the size of the estimates as well as the size of CI.( Moving to a World Beyond “p < 0.05” by Wasserstein RL et al. 2019)

Discussion

The discussion section has a very limited extend and completely lacks a discussion of the findings with related literature. This is extremely important in order to put the findings into a broader scientific content and in order to follow the STROBE statement.

It is not clear which research aim are being discussed. In the first paragraph it is the identification of the 4 trajectories that are (as mentioned above) presented as one of the main results, but this has not previously been described as an aim in the introduction. Neither has the analyses been able to say anything about the attenuation effect of the 3 domains of confounders, separately since they were all adjusted for at one time.

Table 3: I miss a discussion of the precision of the estimates in Table 3, since many are rather unprecise with very large upper confidence limits.

p.36: I do not agree that a correlation of r=0.40 is relatively strong, I find it moderate. I think the choice of a parental reported response should be discussed more into depth, since it is well known from previous literature that there is often a mismatch between children, parents and health professionals' reports of the health of the child and in this case, as mentioned, the 'child' is a grown-up.

**Reviewer 2**

The main concern at the moment is completeness of reporting. The information on all the regression models is relatively sparse. The total variance explained in the models, and added variance at individual steps should be detailed in supplementary analyses. The univariate relationships should also be reported with effect sizes, as this will help understand the relative effect of individual variables in the variance of the outcomes. At present, the focus is on the method (the latent class growth models), while the main aim of the paper is to examine their relationship with outcomes.  

The other main question is why path diagrams were not considered for different outcomes? E.g., IQ may have an impact on externalising trajectories, which may then have an impact on substance use and then unemployment. This effect is lost in the ‘flat’ analysis using logistic regressions.

The value and validity of the Poisson regression is unclear. The number of outcomes is not likely to be a simple count as they are overlapping and not independent of each other. The results from this regression also masks the differential impact of externalising and internalising variables on outcomes. All classes seem to have a similar impact on the count outcome (in Poisson regressions and with overly generalised outcomes such as multiple outcomes, overall poor physical health, any serious physical health event etc) while the logistic regressions on individual outcomes indicate a differential effect consistent with the literature: externalising problems preferentially contributing to low educational attainment, delinquency and substance use and internalising problems preferentially contribute to mental ill health, suicidality, receipt of benefits, poor sleep etc. This is also conspicuously absent from the discussion, especially as such differential impacts would be the main aim of parsing out different data-driven classes of symptom trajectories. The discussion should be more nuanced than what it is now.

From this, what is missing in the manuscript is a more nuanced exploration of trajectories and outcomes- which seems tantalisingly beneath the level of reported data. From this perspective, I strongly recommend adding the 5th and 6th classes into the results. While the authors desire for parsimony is understandable, the 5th and 6th classes seem able to be included based on the entropy. The notion of decreasing moderate trajectories of either type may well be meaningful additions, especially if these do not have the significantly different effects on functioning that the more extreme groups have. As the authors note, reducing the variability in adolescent development to four classes may be overly reductionistic.

We would appreciate receiving your revised manuscript by Jan 02 2020 11:59PM. To enhance the reproducibility of your results, we recommend that if applicable you deposit your laboratory protocols in protocols.io, where a protocol can be assigned its own identifier (DOI) such that it can be cited independently in the future. For instructions see: http://journals.plos.org/plosone/s/submission-guidelines#loc-laboratory-protocols

We look forward to receiving your revised manuscript.

Kind regards,

Monica Uddin, PhD

Academic Editor

PLOS ONE

Journal Requirements:

Reviewers' comments:

Reviewer's Responses to Questions

**Comments to the Author**

1. Is the manuscript technically sound, and do the data support the conclusions?

Reviewer #1: Yes

Reviewer #2: Yes

2. Has the statistical analysis been performed appropriately and rigorously? 

Reviewer #1: I Don't Know

Reviewer #2: Yes

3. Have the authors made all data underlying the findings in their manuscript fully available?

Reviewer #1: Yes

Reviewer #2: No

4. Is the manuscript presented in an intelligible fashion and written in standard English?

Reviewer #1: No

Reviewer #2: Yes

5. Review Comments to the Author

Reviewer #1: This paper aims to investigate the extend to which associations of adolescent mental health trajectories with early-adult functioning were attenuated by adjustment for 1) the effects of childhood functioning, 2) confounding factors and 3) current mental health. However the study does not adequately justify the rationale of its research question although it focus on an important topic. It is really unclear in the current version what new knowledge can be added to this topic and what is the advantages of this study compared to the existing publications.

This together with several shortcomings with the manuscript as mentioned in greater detail below make me unable to recommend this paper for publication as currently presented.

General major comments

1. There is no consistency regarding page numbers throughout the paper, e.g. there are several pages 1 and 2 and at some time the page number jumps from 2 to 34, which makes it difficult to provide specific amendments.

2. The main Aim of the paper is not clear and consistently described throughout the paper. In the abstract the aim is describes as to investigate if there is an association between adolescent mental health trajectories and early-adult functioning. In the introduction the aim focuses on the attenuating separate effects of three domains of confounders. Third, in the beginning of the Discussion section the following is written: "In agreement with out hypothesis, we identified a normative trajectory class with decreasing-low INT and EXT symptoms…..", but no specific hypothesis regarding the mental health trajectories has been formulated?.

3. The arguments for choosing the 17 outcomes are very limited. In p.7 L 166 is written: ' A broad range of early-adulthood outcomes was identified'. A much more detailed description of the term 'functional outcome' is needed, as well as arguments for the choice of outcomes based on a literature review and an explanation of how the individual outcomes are generated and on which basis the thresholds were chosen.

4. The degree of description of the different variables varies a lot. Some of the exposure variables are described in details, whereas the outcome variables are only briefly described.

Major comments

Abstract

The background only contains the aim of the study. A few sentences of background should be added, stating the importance of this study. More compelling is the fact that the aim is not identical with the aim described in the end of the Introduction section or the Discussion section, as mentioned above.

Introduction

In the end of the first paragraph of the Introduction INT and EXT problems are mentioned as part of the aim, before the concepts are explained. This section could be deleated since the aim is already stated in the last paragraph of the Introduction.

Methods

The Sample and procedure section could use a few lines describing the sampling procedures of the Trails. How participants were selected and contacted (e-mail, post?)

p.6, L 128 & : describe the construction of the INT and EXT scales and outcomes in more details. Number of items? Range? scores?

Results

I would recommend not testing for or indicating statistical significance by *. It is only relevant to present and discuss the size of the estimates as well as the size of CI.( Moving to a World Beyond “p < 0.05” by Wasserstein RL et al. 2019)

Discussion

The discussion section has a very limited extend and completely lacks a discussion of the findings with related literature. This is extremely important in order to put the findings into a broader scientific content and in order to follow the STROBE statement.

It is not clear which research aim are being discussed. In the first paragraph it is the identification of the 4 trajectories that are (as mentioned above) presented as one of the main results, but this has not previously been described as an aim in the introduction. Neither has the analyses been able to say anything about the attenuation effect of the 3 domains of confounders, separately since they were all adjusted for at one time.

Table 3: I miss a discussion of the precision of the estimates in Table 3, since many are rather unprecise with very large upper confidence limits.

p.36: I do not agree that a correlation of r=0.40 is relatively strong, I find it moderate. I think the choice of a parental reported response should be discussed more into dept, since it is well known from previous literature that there is often a mismatch between children, parents and health professionals' reports of the health of the child and in this case, as mentioned, the 'child' is a grown-up.

Minor comments

Throughout the paper, do not use the term 'effect' but rater 'association' or other less causal expressions.

Introduction

1. paragraph: There miss half a bracket in: (e.g. (4-7). Change 'start into' to ' transition to' and use another term than 'affected' in the sentence: 'for affected individuals'.

Methods

p.6.L 137-139: is difficult to understand since 'early adult outcome' has not been explained yet. Move to after the description of the outcome.

p.7 L 166: write 'was' instead of 'were'.

p.7 L 147.: second word 'social' should be written with capital letter.

p. 7 L 150 & Table 2: different terms are used: 'Family stress' and 'Parenting stress'

Results

The first section about missing data should be moved to the methods section.

s.3 L251: it is not clear to me how the trajectories of the 4-class models are to be seen in Figure 2?

Reviewer #2: This is a very interesting exploration of the trajectories of mental health symptoms in adolescence and their association with functional predictors. The use of parallel process Latent class growth analyses to model trajectories of symptoms and the relationships with functional outcomes is clinically meaningful.

The main concern at the moment is completeness of reporting. The information on all the regression models is relatively sparse. The total variance explained in the models, and added variance at individual steps should be detailed in supplementary analyses. The univariate relationships should also be reported with effect sizes, as this will help understand the relative effect of individual variables in the variance of the outcomes. At present, the focus is on the method (the latent class growth models), while the main aim of the paper is to examine their relationship with outcomes.

The other main question is why path diagrams were not considered for different outcomes? E.g., IQ may have an impact on externalising trajectories, which may then have an impact on substance use and then unemployment. This effect is lost in the ‘flat’ analysis using logistic regressions.

The value and validity of the Poisson regression is unclear. The number of outcomes is not likely to be a simple count as they are overlapping and not independent of each other. The results from this regression also masks the differential impact of externalising and internalising variables on outcomes. All classes seem to have a similar impact on the count outcome (in Poisson regressions and with overly generalised outcomes such as multiple outcomes, overall poor physical health, any serious physical health event etc) while the logistic regressions on individual outcomes indicate a differential effect consistent with the literature: externalising problems preferentially contributing to low educational attainment, delinquency and substance use and internalising problems preferentially contribute to mental ill health, suicidality, receipt of benefits, poor sleep etc. This is also conspicuously absent from the discussion, especially as such differential impacts would be the main aim of parsing out different data-driven classes of symptom trajectories. The discussion should be more nuanced than what it is now.

From this, what is missing in the manuscript is a more nuanced exploration of trajectories and outcomes- which seems tantalisingly beneath the level of reported data. From this perspective, I strongly recommend adding the 5th and 6th classes into the results. While the authors desire for parsimony is understandable, the 5th and 6th classes seem able to be included based on the entropy. The notion of decreasing moderate trajectories of either type may well be meaningful additions, especially if these do not have the significantly different effects on functioning that the more extreme groups have. As the authors note, reducing the variability in adolescent development to four classes may be overly reductionistic.

6. PLOS authors have the option to publish the peer review history of their article (what does this mean?). If published, this will include your full peer review and any attached files.

Reviewer #1: No

Reviewer #2: No

---

## [Author Response · Author response to Decision Letter 0]

25 Mar 2020

Reviewer 1

General major comments

1. There is no consistency regarding page numbers throughout the paper, e.g. there are several pages 1 and 2 and at some time the page number jumps from 2 to 34, which makes it difficult to provide specific amendments.

We thank the reviewer for pointing out this error. We have adjusted the page numbering. 

2. The main Aim of the paper is not clear and consistently described throughout the paper. In the abstract the aim is describes as to investigate if there is an association between adolescent mental health trajectories and early-adult functioning. In the introduction the aim focuses on the attenuating separate effects of three domains of confounders. Third, in the beginning of the Discussion section the following is written: "In agreement with our hypothesis, we identified a normative trajectory class with decreasing-low INT and EXT symptoms…..", but no specific hypothesis regarding the mental health trajectories has been formulated?.

We agree with the reviewer that our aims and hypothesis were not clearly and consistently described. Overall, our study has two aims: First, to identify distinct developmental trajectories of (co-occurring) internalizing and externalizing symptoms during adolescence, and (2) do examine the associations between these trajectories and early-adult functioning unadjusted and adjusted for childhood functioning, confounders and early-adult mental health status. 

In the revised manuscript, we made sure to be consistent when describing these two aims. The aims are reported as follows:

Abstract (p.2): “The aim of this study was twofold: (1) to identify distinct developmental trajectories of (co-occurring) internalizing and externalizing mental health symptoms over the course of adolescence (ages 11-19), and (2) to document the associations between these adolescent trajectories and economic, social, and health outcomes in young adulthood (age 22), unadjusted and adjusted for childhood functioning, putative confounders and current mental health.”

Introduction (p. 7-8, L142-150): “In sum, to better understand the links between adolescent mental health and early-adult functioning, the aim of this study was twofold. First, we aimed to identify distinct developmental trajectories of (co-occurring) INT and EXT symptoms over the course of adolescence. Second, we aimed to document the associations between these adolescent trajectories and functional outcomes in young adulthood in three domains: economic, social, and health, unadjusted and adjusted for childhood functioning, putative confounders and current mental health. To avoid operational confounding between adolescent trajectories and early-adult functional outcomes, the time frame of the trajectories was limited to adolescence (ages 11-19), whereas the functional outcomes were assessed at age 22 and typically referred to the preceding month(s).”

Discussion (p. 34, L446-450): “The aims of this prospective study were (1) to identify distinct developmental trajectories of (co-occurring) internalizing and externalizing mental health symptoms over the course of adolescence (ages 11-19), and (2) to document the associations between these adolescent trajectories and economic, social, and health outcomes in young adulthood (age 22), unadjusted and adjusted for childhood functioning, putative confounders and current mental health.

Our hypotheses were also added to the introduction (P. 34, L151-156): “Based on previous literature, we expected to identify a normative trajectory class with low INT and EXT symptoms, a high-severity trajectory class with high INT and EXT symptoms, and a moderate-severity class with predominantly INT symptoms. Moreover, we expected that the most severe trajectory class would be associated with the poorest outcomes and that these associations would attenuate, but remain significant, after adjustment for childhood functioning, confounding factors and current mental health.”

3. The arguments for choosing the 17 outcomes are very limited. In p.7 L 166 is written: ' A broad range of early-adulthood outcomes was identified'. A much more detailed description of the term 'functional outcome' is needed, as well as arguments for the choice of outcomes based on a literature review and an explanation of how the individual outcomes are generated and on which basis the thresholds were chosen.

In the revised manuscript, we have added to the description of the term ‘functional outcome’, explained our choice of outcomes, how they were generated and on which basis the for both the age 22 and age 11 functioning variables. The revised text is now as follows (p. 10, L199-213): “Our aim was to identify a broad range of early-adulthood outcomes that typically impede functioning for most individuals. The choice of outcomes was primarily based on the Copeland et al. studies and our 2017 study linking childhood and adolescence psychiatric history on early-adult functioning [13]. There were some adaptations. For example, contrary to Copeland et al. [9], we did not have data on serious criminality and incarceration for all participants, but instead included substantial antisocial behavior as outcome measure. The selected outcomes covered the economic, social and health domains and were mostly generated through self-report. Table 1 describes the 17 outcomes assessed at age 22 (wave 5), on average 38 months after the wave 4 ASR (age 19y). Most outcomes refer to the 3-6 months preceding wave 5, some to the wave 4-5 interval. In some cases, the thresholds for outcomes were self-evident (e.g., early parenthood yes/no, receiving social benefits yes/no). Where such thresholds were not available (i.e., ordinal or dimensional measures), outcomes were defined to identify approximately 10% of participants or fewer (e.g., delinquency was defined as 4+ self-reported delinquent acts, problematic alcohol use was defined as a score ≥ 14 on the alcohol use disorder identification test) to ensure the outcome was deviant. Additional information about outcomes is available in Table 1.” 

.”

4. The degree of description of the different variables varies a lot. Some of the exposure variables are described in details, whereas the outcome variables are only briefly described.

We thank the reviewer for pointing this out. Indeed, the length of the descriptions in the text differs for the outcome variables versus the exposure variables. This is because the outcome variables were already described in Table 1 and repeating this information in the text felt a bit redundant. However, we agree that we should provide more information on how exactly the outcome measures were generated (i.e., which instruments were used for the age 22 and equivalent age 11 childhood functioning variables, and the thresholds for deviance). We have now included this information in the revised Table 1, including literature references. 

Major comments

Abstract

5. The background only contains the aim of the study. A few sentences of background should be added, stating the importance of this study. More compelling is the fact that the aim is not identical with the aim described in the end of the Introduction section or the Discussion section, as mentioned above.

We have added a few lines on the background of the study to the revised abstract. We also thank the reviewer for pointing out that the aim as described in the abstract did not match the aims as mentioned in the introduction and discussion session. In the revised manuscript, we now describe our two aims more clearly and consistently. 

The new background section of the revised abstract is as follows (P2, L. 14-22): “Mental health problems during adolescence may create a problematic start into adulthood for affected individuals. Usually, categorical indicators of adolescent mental health issues (yes/no psychiatric disorder) are used in studies into long-term functional outcomes. This however does not take into account the full spectrum of mental health, nor does it consider the trajectory of mental health problem development over time. The aim of this study was twofold: (1) to identify distinct developmental trajectories of (co-occurring) internalizing and externalizing mental health symptoms over the course of adolescence (ages 11-19), and (2) to document the associations between these adolescent trajectories and economic, social, and health outcomes in young adulthood (age 22), unadjusted and adjusted for childhood functioning, putative confounders and current mental health.”

Introduction

6. In the end of the first paragraph of the Introduction INT and EXT problems are mentioned as part of the aim, before the concepts are explained. This section could be deleted since the aim is already stated in the last paragraph of the Introduction.

We agree and have deleted the last section of the first paragraph as suggested by the reviewer.

Methods

7. The Sample and procedure section could use a few lines describing the sampling procedures of the Trails. How participants were selected and contacted (e-mail, post?)

We have added extra following information on the sampling procedures of TRAILS to the revised manuscript (P.7-9=8, L165-177): “Participants were selected from five municipalities in the North of the Netherlands, both urban and rural, including the three largest cities. Children born between 1 October 1989 and 30 September 1991 were eligible for inclusion, providing they met the inclusion criteria and their schools were willing to participate (21). Of the initially 135 primary schools in the area, encompassing 3483 eligible children, over 90% (n=3145 children), agreed to participate in the study. Then, parents/guardians and children were informed though information brochures about the study goals, selection procedure, confidentiality and measurements. Shortly thereafter, an interviewer contacted the parents by telephone to invite them to participate. Another 210 children were excluded because they were unable or incapable to participate (e.g., due to severe mental retardation or physical illness or handicap or if no Dutch-speaking parent was available, enrolling a total of 2935 eligible children. Through extended efforts (including telephone calls, reminders and home visits), 76% of these children and their parents consented to participate (wave 1, n=2230, mean age=11.1 years, SD=0.6, 50.8% girls).”

8. p.6, L 128 & : describe the construction of the INT and EXT scales and outcomes in more details. Number of items? Range? scores?

As suggested by the reviewer, we have added more information about the construction of the INT and EXT scales to the text (P. 9, L186-196): “Mental health symptoms were assessed with the INT and EXT scales of the Youth Self Report (YSR) at waves 1-3 and the Adult Self Report (ASR) at wave 4 [26,27]. Participants responded to each item on a three-point scale (0 = not true, 1 = somewhat true, 2 = very often or always true). Both YSR and ASR have good validity and reliability [28,29], which were also confirmed for the Dutch versions [30]. The INT scale consists of the subscales Anxious/Depressed, Withdrawn/Depressed and Somatic Complaints (31 items for the YSR [range 0-52], 39 items for the ASR [range 0-78]). The EXT scale consists of the subscales Rule-Breaking Behaviour, Aggressive Behaviour and Intrusive (32 items for the YSR [range 0-54], 35 items for the ASR [range 0-70]). Internal consistency for the INT and EXT subscales was good to excellent across all waves (T1: �INT=0.87, �EXT=0.85; T2: �INT=0.88, �EXT=0.85; T3: �INT=0.89, �EXT=0.87; T4: �INT=0.93, �EXT=0.89.”

We have added more information about the construction of the outcomes variables (age 22 and equivalent age 11 childhood functioning) in Table 1, See also our response to comment 4. 

Results

9. I would recommend not testing for or indicating statistical significance by *. It is only relevant to present and discuss the size of the estimates as well as the size of CI.( Moving to a World Beyond “p < 0.05” by Wasserstein RL et al. 2019)

We thank the reviewer for pointing us to this debate and, by extension, the ASA statement on p-values and statistical significance (Wasserstein & Lazar, 2016). Indeed, conclusions shouldn’t be based solely on whether an association or effect was found to be statistically significant (as indexed by p=<.05), this is why, in the initial submission, we already reported and discussed the 95% confidence intervals (CI) of the odds ratio’s. However, following the reviewers suggestion, in the revised manuscript, we also removed the indication of statistical significant by asterisks and the bold print. In the revised text, we now refrain from using the terms ‘statistically significant’ and ‘significantly predicted’.

Discussion

10. The discussion section has a very limited extend and completely lacks a discussion of the findings with related literature. This is extremely important in order to put the findings into a broader scientific content and in order to follow the STROBE statement.

We thank the reviewer for pointing this out and agree that the discussion lacks a nuanced discussion of the finding, especially concerning the added value of the trajectories and the differential findings for externalizing and internalizing symptoms. 

In the revised manuscript, we have profoundly rewritten the discussion section. We have extended the discussion of the implications of our trajectory findings, we have added an extra paragraph discussing the differential associations for internalizing and externalizing trajectories, we have expanded the limitation section to include a discussion on using parent-reported current mental health and the large CI’s for some of the outcomes, we have added an extra paragraph on recommendations for future research, and have extended the concluding comments section. 

11. It is not clear which research aim are being discussed. In the first paragraph it is the identification of the 4 trajectories that are (as mentioned above) presented as one of the main results, but this has not previously been described as an aim in the introduction. Neither has the analyses been able to say anything about the attenuation effect of the 3 domains of confounders, separately since they were all adjusted for at one time.

As previously indicated, our aims were not clearly and consistently described throughout the paper. Our two main research aims were: (1) to identify distinct trajectories of (co-occurring) internalizing and externalizing symptoms over the course of adolescence, and (2) to examine the associations between these trajectories and early-adult functioning, unadjusted and adjusted for childhood functioning, selected putative confounders and early-adult mental health status. It was not our intention to test the attenuation effects of the three domains of confounders separately, although the way it was described in the text certainly gave that impression. This is, because in earlier versions of the manuscript, we did test for separate attenuation effects for the confounder domains. We found, however, that this did not add much information compared to adjusting for the confounders at one time. Unfortunately, the text was not correctly adjusted. 

In the revised manuscript, we have rewritten this section of the introduction (P. 34, L444-467): “ The aims of this prospective study were (1) to identify distinct developmental trajectories of (co-occurring) internalizing and externalizing mental health symptoms over the course of adolescence (ages 11-19), and (2) to document the associations between these adolescent trajectories and economic, social, and health outcomes in young adulthood (age 22), unadjusted and adjusted for childhood functioning, putative confounders and current mental health. In agreement with our hypothesis, we identified a normative trajectory class with decreasing-low INT and EXT symptoms, a trajectory class with moderate-high INT and EXT symptoms, and a moderate-severity class with predominantly INT symptoms. In addition, we identified a trajectory class with predominantly EXT symptoms that decreased in severity across adolescence. Also, in line with our hypothesis, was that compared to the decreasing-low trajectory, the other three trajectories generally predicted less optimal early-adult outcomes, with the strongest effects observed for individuals with continuous moderate-high levels of both INT and EXT symptoms throughout adolescence. The effects of the other two trajectories varied with outcome. 

The associations between adolescent mental health and early-adult outcome largely remained significant after adjustment for pre-adolescent functioning and selected confounders. This suggest that the observed associations between mental health trajectories during adolescence and early-adult functioning are not explained by other known correlates of functioning, such as childhood functioning and correlates of both psychopathology and functioning, such as sex, IQ, temperament, family SES, parenting stress, and parenting style. Finally, current mental health status accounted for additional variance in early-adult outcomes, but this did not overlap strongly with the trajectories of mental health in adolescence, which remained important as well. This suggests that both past and current mental health are important for understanding functioning in early adolescence.”

12. Table 3: I miss a discussion of the precision of the estimates in Table 3, since many are rather unprecise with very large upper confidence limits.

We have added a discussion of the imprecision of some of the estimates in Table 3 to the limitation section of the discussion (P. 37, L543-545): “(5) Some of the estimates had rather large upper confidence limits, likely due to the low exposure rates of these outcomes within the classes, suggesting that these findings should be interpreted with caution.”

13. p.36: I do not agree that a correlation of r=0.40 is relatively strong, I find it moderate. I think the choice of a parental reported response should be discussed more into depth, since it is well known from previous literature that there is often a mismatch between children, parents and health professionals' reports of the health of the child and in this case, as mentioned, the 'child' is a grown-up.

Low to moderate parent-child agreement is indeed a known problem found in most studies. We chose to use parent-report to avoid shared method variance with the self-reported early-adult outcomes, but acknowledge that parents may not be the best informant in the limitation section of the discussion. Although we agree with the reviewer that the correlation should best be described as moderate, in our opinion the correlation was substantial and similar or larger than typically found at younger ages. 

We have rewritten this section of the limitations (P, 37, L537-543):”Parents may not be the best informant of the mental health status of their 22-year old children, because the child may no longer live at home or have chosen not to disclose mental health issues to their parent(s). Low to moderate parent-child agreement is found in most studies describing discrepancies in reports of mental health disorders [51]. We chose parent-reported current mental health to avoid shared method variance with the self-reported early-adult outcomes. In our sample, the correlation between self- and parent-reported current mental health problems was r ≈.40, which is similar or larger than that typically found at younger ages.”

Reviewer 2

The main concern at the moment is completeness of reporting. The information on all the regression models is relatively sparse. The total variance explained in the models, and added variance at individual steps should be detailed in supplementary analyses. The univariate relationships should also be reported with effect sizes, as this will help understand the relative effect of individual variables in the variance of the outcomes. 

We have added additional information on the regression models to the method section (p. 16-17, L287-298): “Dependent variables were the early-adult outcome variables as described in Table 1. We ran a separate analysis for each outcome. In the unadjusted model, the dependent variables were, separately, regressed on the trajectory dummy variables. The adjusted model included the following covariates: childhood functioning, child covariates, and current INT and EXT symptoms. In the adjusted model, the dependent variables were, separately, regressed on the trajectory dummies and all covariates simultaneously. We used the standardized residual variance of current mental health that we obtained by regressing trajectory class membership on the parent- reported INT and EXT scales in a separate analysis and saving the standardized residuals.”

In addition, we have also added the explained total variance of the unadjusted and adjusted models to Table 3. Following reviewer 1’s suggestion, we omitted indicating statistical significance with asterisks and bold print in Table 3 (please see our response to comment 9 of reviewer 1). It was not the aim of this paper to test the attenuation effects of the three domains of confounders separately, although the way it was described in the text certainly gave that impression. We have adjusted this in the revised manuscript (see also our response to comment 11 of reviewer 1). Because it was not the aim of the paper, we did not include the estimates of each individual covariate to the table, except for current INT and EXT symptoms. 

At present, the focus is on the method (the latent class growth models), while the main aim of the paper is to examine their relationship with outcomes. 

We agree with the reviewer that the focus of the paper was more on the method than on the main aims of the paper. In the revised discussion, we have made an effort to put the findings in a broader scientific context, see also our response to comment 10 of reviewer 1. 

In the revised manuscript, we have profoundly rewritten the discussion section. We have extended the discussion of the implications of our trajectory findings, we have added an extra paragraph discussing the differential associations for internalizing and externalizing trajectories, we have expanded the limitation section to include a discussion on using parent-reported current mental health and the large CI’s for some of the outcomes, we have added an extra paragraph on recommendations for future research, and have extended the concluding comments section. 

The other main question is why path diagrams were not considered for different outcomes? E.g., IQ may have an impact on externalising trajectories, which may then have an impact on substance use and then unemployment. This effect is lost in the ‘flat’ analysis using logistic regressions.

We believe the reviewer raises an interesting point here. Our main aim was to examine the impact of trajectories of mental health symptoms during adolescence on early-adult functioning and we feel that examining specific path diagrams for different outcomes is beyond the scope of the current paper. Although highly interesting, these path analyses would answer entirely different research questions about the mechanisms behind associations between adolescent mental health on early-adult functioning. This would also require a different set-up of the available TRAILS data (e.g., in the example raised by the reviewer, we would need to take into account the temporal order of assessing IQ, EXT trajectory, substance use and unemployment in order to test if one variable predicts another). 

We have added the option of path diagrams to the discussion section as a suggestion for future research into the mechanisms that drive the associations between adolescent mental health issues and early-adult functioning (P. 38, L557-569): “Adding to existing literature on long-term associations between adolescent mental health problems and poor adult functional outcomes, our study has shown that these associations are not limited to individuals with continuously high levels of mental health problems, but also in individuals with moderate symptom trajectories, not attenuated by current mental health symptoms, and that some of the outcomes were differentially associated with distinct symptom trajectories (e.g., problematic substance use problems were only significantly associated with trajectories including elevated EXT symptoms). What we did not address in this study are the potential mechanisms driving the association between adolescent mental health issues and early-adult functional outcomes (that are also likely to be interrelated). For example, IQ may have an impact on externalizing trajectories which may then have an impact on substance use and subsequently unemployment. Examining these underlying pathways would give important insights into the factors that drive the associations and point towards potential targets for prevention and intervention efforts, but it was beyond the scope of the current study.”

The value and validity of the Poisson regression is unclear. The number of outcomes is not likely to be a simple count as they are overlapping and not independent of each other. The results from this regression also masks the differential impact of externalising and internalising variables on outcomes. All classes seem to have a similar impact on the count outcome (in Poisson regressions and with overly generalised outcomes such as multiple outcomes, overall poor physical health, any serious physical health event etc) while the logistic regressions on individual outcomes indicate a differential effect consistent with the literature: externalising problems preferentially contributing to low educational attainment, delinquency and substance use and internalising problems preferentially contribute to mental ill health, suicidality, receipt of benefits, poor sleep etc. 

We agree with the reviewer that the value and validity of the Poisson regression in unclear and decided to drop the count analyses from our paper. The reviewer is right in stating that the number of outcomes is not likely to be a simple count (an oversight from our part) and that the results from these count variables do not really add to the story (all classes seem to have a similar impact on the count outcomes). Therefore, we omitted the count variables from our analyses and changed the text and tables accordingly. 

We decided, following the approach of Copeland et al. (2015), to included two variables: ‘any’ and ‘multiple’ in and compared the classes pairwise on these variables (descriptive analysis only) (P. 16, L285-287): “The classes were pairwise compared on descriptives and covariates, and on the percentage of individuals reported any and multiple (2+) poor outcomes”.

Results were reported and presented in Figure 2 (P. 23,, L409-415): “Figure 2 presents the distribution of any and multiple poor outcomes across the four trajectory classes. Finding showed that of the individuals with decreasing-low INT and EXT symptoms throughout adolescence, 45.0% reported a poor adult outcome and 20% reported two or more poor outcomes. In comparison, of the individuals with continuously moderate-high levels of both INT and EXT symptoms, 84.6% reported a poor outcome and 62.8% reported two or more. Of the two other classes, around 70% of the individuals reported one poor outcome and around 40% two or more poor outcomes.”

And discussed in the implications section of the revised discussion (P. 35, L471-506): “Taken together [...] Close to two thirds of these individuals reported two or more adverse outcomes, compared to one in five out of individuals with low levels of INT and EXT symptoms. […] From these two trajectory classes, around 40% reported two or more poor outcomes. […] Whereas youth with persistently high problem levels had a three-fold increased risk multiple outcomes compared to youth with persistently low problem levels (62.8% versus 20%), over one third of them had fewer than two poor outcomes.”

This is also conspicuously absent from the discussion, especially as such differential impacts would be the main aim of parsing out different data-driven classes of symptom trajectories. The discussion should be more nuanced than what it is now.

We agree with the reviewer that the discussion lacked a nuanced discussion on the differential impacts of distinct symptom trajectories. In the revised discussion, we have made an effort to put the findings in a broader scientific context, see also our response to comment 10 of reviewer 1. 

In the revised manuscript, we have profoundly rewritten the discussion section. We have extended the discussion of the implications of our trajectory findings, we have added an extra paragraph discussing the differential associations for internalizing and externalizing trajectories, we have expanded the limitation section to include a discussion on using parent-reported current mental health and the large CI’s for some of the outcomes, we have added an extra paragraph on recommendations for future research, and have extended the concluding comments section. 

From this, what is missing in the manuscript is a more nuanced exploration of trajectories and outcomes- which seems tantalisingly beneath the level of reported data. From this perspective, I strongly recommend adding the 5th and 6th classes into the results. While the authors desire for parsimony is understandable, the 5th and 6th classes seem able to be included based on the entropy. The notion of decreasing moderate trajectories of either type may well be meaningful additions, especially if these do not have the significantly different effects on functioning that the more extreme groups have. As the authors note, reducing the variability in adolescent development to four classes may be overly reductionistic.

We completely understand the reviewer’s point of view. We actually debated about which model to choose for further analyses ourselves, since the fit indices gave no definitive clues as to which model was most optimal. Therefore, model interpretability was considered to select the most optimal solution. The 4-, 5- and 6-class models showed at least two codevelopment classes that were characterized by INT and EXT symptoms that did not change in tandem with each other, making them conceptually more interesting than the 2-, and 3-class models. The 7- and 8-class models were not further considered due to small class-sizes (at least one class with n<10%) and estimation problems. The 4-class model showed four qualitatively distinct trajectory classes, with the ‘healthiest’ class comprising 30% of the sample. The 5-class model added an in-between group with a similar decreasing-severity more EXT than INT trajectory, but with low to moderate overall levels of EXT and INT symptoms. The majority of individuals from the ‘healthiest’ class in the 4-class model were transferred to this new in-between class, leaving the ‘healthiest’ class to comprise of only 12% of the sample. The 6-class model showed similar trajectory classes as the 5-class model, but added an additional increasing-severity trajectory class with more severe EXT than INT. Also in this model, the healthiest class comprised 12% of the sample. We decided not to consider the 5-class model because it did not reveal an additional qualitatively distinct class compared to 4-class model. This was however the case for the 6-class model, leaving both the 4- and 6-class models as viable options. 

We debated about choosing the more parsimonious 4-class model, with the risk of being overly reductionistic in reducing the variability in adolescent development to four classes as the reviewer already mentioned. Or choosing the 6-class model with the additional trajectory class, with the risk of running into estimations problems due to the much smaller reference group (12 vs. 30% of the sample) combined with low exposure rates for some of the outcomes (e.g., suicidality, serious physical event, teen pregnancy). In addition, choosing the 6-class model also would include multiple comparisons between quantitively but not qualitatively distinct classes that are unlikely to reveal significantly different effects on functioning. In the end, we opted for the more parsimonious 4-class model for further analyses. This would also make interpretation of between trajectory differences more easily understandable.

We have added this decision process to the main text of the revised manuscript (p. 17-19, L317-369): “ Table 2 provides the fit indices for the PP-LCGA class-solutions. In the PP-LCGA, the AIC and BIC decreased and the BLRT remained significant (p<.0001) with each class addition, giving no definitive clues as to which model was most optimal. Therefore, model interpretability of the models was considered to select the most optimal solution. The 4-, 5- and 6-class models showed at least two codevelopment classes that were characterized by INT and EXT symptoms that did not change in tandem with each other, making them conceptually more interesting than the 2-, and 3-class models. The 7- and 8-class models were not further considered due to small class-sizes (at least one class with n<10%) and estimation problems.

The 4-class model showed a high-severity trajectory class with individuals displaying continuously high INT and EXT symptoms throughout adolescence (‘Continuous moderate-high INT + EXT’; n=298), a low-severity trajectory class with individuals with decreasing low INT and EXT symptoms throughout adolescence (‘decreasing-low INT + EXT’; n=460; the healthiest and thus reference class, 30% of the sample), a moderate-severity trajectory class with more severe INT than EXT symptoms over time (‘more INT than EXT-moderate’; n=414), and a decreasing-severity trajectory class with more severe EXT than INT symptoms over time (‘more EXT than INT-decreasing’; n=352). The 5-class model showed similar classes but added an in-between group with a similar decreasing-severity more EXT than INT trajectory, but with low to moderate overall levels of EXT and INT symptoms. The majority of individuals from the decreasing low (reference) class in the 4-class model were transferred to this new in-between class, leaving a reference class of n=187 (12% of total sample). The 6-class model showed similar trajectory classes as the 5-class model, but added an additional increasing-severity trajectory class with more severe EXT than INT. Also in this model, the healthiest class comprised 12% of the sample (n=189). The 5-class model was not further considered since no qualitatively distinct class emerged in this model compared to 4-class model. This left the 4- and 6-class models as viable options. We debated about choosing the more parsimonious 4-class model, with the risk of being overly reductionistic in reducing the variability in adolescent development to four classes. Or choosing the more fine-grained 6-class model, with the risk of estimations problems due to the much smaller reference group combined with low exposure rates for some outcomes, and in addition, multiple comparisons between quantitively but not qualitatively distinct classes that are unlikely to reveal significantly different effects on functioning. In the end, the more parsimonious 4-class model was chosen over the 6-class model for further analyses, based on the substantive meaningfulness and interpretability of the classes.”.

However, because the additional increasing, EXT>INT class from the 6-class model has the potential off differential associations with functional outcomes compared to the other, qualitatively distinct, trajectories from the 4-class model, we decided to run the analyses on the 6-classes and document their associated functional outcomes (findings of the other trajectories are not presented in the main text, but can be found in the Supplemental material S1 Fig and S1 Table) (P. 19, L.370-374).

As expected, we did run into some estimation problems with very imprecise estimations for some of the outcomes (95% confidence intervals ranging from 4.70-293.39 e.g. for loneliness in the moderate-stable INT+EXT class compared to the reference group), probably due to low exposure rates within the much smaller reference group (see S1 Table). 

The additional increasing EXT>INT class was at an increased risk of poor economic outcomes, social difficulties, delinquency, problematic substance use, mental health concerns and poor subjective health. This set of associated functional outcomes was very similar to the moderate-high INT+EXT group from the 4-class model (P. 24, L434-436)

We discussed this in the revised manuscript (P. 36, L499-505): “This additional increasing EXT>INT class was at similar increased risk of the continuous moderate, INT+EXT class, despite lower INT levels and much lower EXT symptom levels in early adolescence, the associated outcomes hardly differed. This suggest that predominantly the presence of EXT symptoms, particularly during late adolescence, may drive poor early-adult functioning. However, these findings should be interpreted with caution given that many estimates had rather large confidence intervals, possibly due to the much smaller reference sample for the increasing EXT>INT class.”

Journal Requirements:

Thank you for pointing this out to us. We have made sure our manuscript meets PLOS ONE’s style requirements.

We have included captions for our Supporting information files at the end of our manuscript.

An updated version of our data availability statement was provided.

---

## [Editor Report · Decision Letter 1]

11 May 2020

The association of developmental trajectories of adolescent mental health with early-adult functioning

PONE-D-19-24355R1

Dear Dr. Oerlemans,

We are pleased to inform you that your manuscript has been judged scientifically suitable for publication and will be formally accepted for publication once it complies with all outstanding technical requirements.

With kind regards,

Geilson Lima Santana, M.D., Ph.D.

Academic Editor

PLOS ONE
---

## [Editor Report · Acceptance letter]

19 May 2020

PONE-D-19-24355R1 

The association of developmental trajectories of adolescent mental health with early-adult functioning 

Dear Dr. Oerlemans:

I am pleased to inform you that your manuscript has been deemed suitable for publication in PLOS ONE. Congratulations! Your manuscript is now with our production department. 

With kind regards,

on behalf of

Dr. Geilson Lima Santana 

Academic Editor

PLOS ONE